# Sulfonylurea Receptor 1 in Central Nervous System Injury: An Updated Review

**DOI:** 10.3390/ijms222111899

**Published:** 2021-11-02

**Authors:** Ruchira M. Jha, Anupama Rani, Shashvat M. Desai, Sudhanshu Raikwar, Sandra Mihaljevic, Amanda Munoz-Casabella, Patrick M. Kochanek, Joshua Catapano, Ethan Winkler, Giuseppe Citerio, J. Claude Hemphill, W. Taylor Kimberly, Raj Narayan, Juan Sahuquillo, Kevin N. Sheth, J. Marc Simard

**Affiliations:** 1Department of Neurology, Barrow Neurological Institute and St. Joseph’s Hospital and Medical Center, Phoenix, AZ 85013, USA; ruchiramjha@gmail.com (R.M.J.); shashvatdesaimd@gmail.com (S.M.D.); 2Department of Translational Neuroscience, Barrow Neurological Institute and St. Joseph’s Hospital and Medical Center, Phoenix, AZ 85013, USA; Anupama.Rani@Barrowneuro.org (A.R.); Sudhanshu.Raikwar@Barrowneuro.org (S.R.); sandra.mihaljevic@Barrowneuro.org (S.M.); Amanda.Munoz-Casabella@Barrowneuro.org (A.M.-C.); 3Department of Neurosurgery, Barrow Neurological Institute and St. Joseph’s Hospital and Medical Center, Phoenix, AZ 85013, USA; joshua.catapano@barrowbrainandspine.com (J.C.); Ethan.Winkler@barrowbrainandspine.com (E.W.); 4Clinical and Translational Science Institute, School of Medicine, University of Pittsburgh, Pittsburgh, PA 15213, USA; kochanekpm@pitt.edu; 5Department of Critical Care Medicine, University of Pittsburgh, Pittsburgh, PA 15213, USA; 6Department of Pediatrics, School of Medicine, University of Pittsburgh, Pittsburgh, PA 15213, USA; 7Safar Center for Resuscitation Research, School of Medicine, University of Pittsburgh, Pittsburgh, PA 15213, USA; 8School of Medicine and Surgery, University of Milan-Bicocca, 20126 Milan, Italy; giuseppe.citerio@unimib.it; 9Neurointensive Care Unit, Department of Neuroscience, San Gerardo Hospital, ASST—Monza, 20900 Monza, Italy; 10Department of Neurology, University of California, San Francisco, CA 94143, USA; claude.hemphill@ucsf.edu; 11Division of Neurocritical Care and Center for Genomic Medicine, Department of Neurology, Massachusetts General Hospital, Boston, MA 02114, USA; WTKIMBERLY@mgh.harvard.edu; 12Department of Neurosurgery, Donald and Barbara Zucker School of Medicine at Hofstra/Northwell, North Shore University Hospital, Manhasset, NY 11549, USA; RNarayan@northwell.edu; 13Neurotrauma and Neurosurgery Research Unit (UNINN), Vall d’Hebron Research Institute (VHIR), 08035 Barcelona, Spain; sahuquillo.juan@gmail.com; 14Neurotraumatology and Neurosurgery Research Unit, Universitat Autònoma de Barcelona (UAB), 08193 Barcelona, Spain; 15Department of Neurosurgery, Vall d’Hebron University Hospital, 08035 Barcelona, Spain; 16Division of Neurocritical Care and Emergency Neurology, Department of Neurology, School of Medicine, Yale University, New Haven, CT 06510, USA; kevin.sheth@yale.edu; 17Department of Neurosurgery, University of Maryland School of Medicine, Baltimore, MD 21201, USA; 18Department of Pathology, University of Maryland School of Medicine, Baltimore, MD 21201, USA; 19Department of Physiology, University of Maryland School of Medicine, Baltimore, MD 21201, USA

**Keywords:** sulfonylurea receptor, edema, cellular swelling, traumatic brain injury, stroke, SUR 1, TRPM4, clinical trials

## Abstract

Sulfonylurea receptor 1 (SUR1) is a member of the adenosine triphosphate (ATP)-binding cassette (ABC) protein superfamily, encoded by Abcc8, and is recognized as a key mediator of central nervous system (CNS) cellular swelling via the transient receptor potential melastatin 4 (TRPM4) channel. Discovered approximately 20 years ago, this channel is normally absent in the CNS but is transcriptionally upregulated after CNS injury. A comprehensive review on the pathophysiology and role of SUR1 in the CNS was published in 2012. Since then, the breadth and depth of understanding of the involvement of this channel in secondary injury has undergone exponential growth: SUR1-TRPM4 inhibition has been shown to decrease cerebral edema and hemorrhage progression in multiple preclinical models as well as in early clinical studies across a range of CNS diseases including ischemic stroke, traumatic brain injury, cardiac arrest, subarachnoid hemorrhage, spinal cord injury, intracerebral hemorrhage, multiple sclerosis, encephalitis, neuromalignancies, pain, liver failure, status epilepticus, retinopathies and HIV-associated neurocognitive disorder. Given these substantial developments, combined with the timeliness of ongoing clinical trials of SUR1 inhibition, now, another decade later, we review advances pertaining to SUR1-TRPM4 pathobiology in this spectrum of CNS disease—providing an overview of the journey from patch-clamp experiments to phase III trials.

## 1. Introduction

Sulfonylurea receptor 1 (SUR1) is a member of the adenosine triphosphate (ATP)-binding cassette (ABC) protein superfamily, which encompasses a large group of membrane proteins that assist in and regulate the transport of ions and molecules across lipid bilayers [1,2]. To mediate this action, ABC proteins use energy derived from ATP hydrolysis. The ABC transporter subfamily encoded by the *Abcc* genes consists of three classes: multidrug resistance-associated proteins encoded by *Abcc1–6* and *Abcc10–13*, the cystic fibrosis conductance regulator encoded by *Abcc7* and the sulfonylurea receptors (SUR) encoded by *Abcc8* and *Abcc9* [3]. The SUR proteins appear in several isoforms with variable tissue-predominant expression patterns: for example, SUR1 is expressed in pancreatic β-cells and neurons, SUR2A is expressed in cardiac and skeletal muscle, and SUR2B is expressed in vascular smooth muscle [4].

Discovered approximately two decades ago, an octameric channel regulated by SUR1 is now increasingly recognized as a key mediator of central nervous system (CNS) cellular swelling via the association of four SUR1 subunits with four subunits of an ATP and calcium-sensitive non-selective cation pore-forming subunit, formerly known as NC_Ca-ATP_ and subsequently identified as transient receptor potential melastatin 4 (TRPM4) [5,6,7,8]. The SUR1-TRPM4 channel, when open, results in sodium influx, cell depolarization, intracellular edema, and, ultimately, oncotic cell death [5,6,9,10,11,12]. This channel is not present in the normal CNS but is transcriptionally upregulated after various forms of CNS injury, and contributes to cerebral edema.

Cerebral edema, an accumulation of intracerebral fluid that increases the net brain-tissue water mass, is a key detrimental hallmark of various CNS injuries including ischemic stroke, traumatic brain injury (TBI), subarachnoid hemorrhage (SAH), intracerebral hemorrhage (ICH), cardiac arrest (CA), spinal cord injury (SCI), intracranial tumors, infections, autoimmune processes, and systemic conditions such as liver failure and sepsis [12,13,14]. In the context of a rigid skull, by increasing brain-tissue volume, cerebral edema often lowers brain-tissue perfusion, thereby causing cellular distress/injury. This also increases the risk of herniation and death [10]. This relationship between intracranial pressure (ICP) and cerebral edema was first described in 1783 by the Scottish Surgeon Alexander Monro, and is now well known as the Monro–Kellie hypothesis [15]. The increased pressure can also limit cerebral perfusion, and further perpetuate secondary injury. Mechanistically, oncotic edema of endothelial cells and breakdown of the blood–brain barrier (BBB) further contribute to other forms of secondary injury such as hemorrhage progression [16]. Currently, cerebral edema and resultant intracranial hypertension are treated with non-specific therapies such as hyperosmolar agents and, when severe, decompressive craniectomy. Although these therapies can be life-saving, they do not always improve functional outcomes, nor do they target or prevent the underlying pathobiological mechanisms [17,18,19,20]. Advances in precision medicine have led to an increasing recognition of the fact that a ‘one-size -fits-all’ approach is likely suboptimal [20,21,22,23], and the potential benefits of a targeted approach.

SUR1-TRPM4 expression and contribution to cerebral edema and secondary injury has been demonstrated across several types of CNS injury. SUR1-TRPM4 inhibition decreases cerebral edema and hemorrhage progression in multiple preclinical models, as well as some early clinical studies across a spectrum of CNS injury [9,11,24,25,26,27,28,29,30,31,32,33,34,35,36,37,38,39,40,41,42,43,44,45,46,47,48,49,50]. These studies underscore its potential as a translatable therapeutic target and biomarker after CNS injury. A comprehensive review on the pathophysiology and role of SUR1 in the CNS was published in 2012, approximately 10 years after the channel’s discovery [6]. Now, another decade later, we review advances in the depth and breadth of understanding of the role of SUR1-TRPM4 in CNS disease and provide an overview of the journey from patch-clamp experiments to phase III clinical trials. Here, we outline the reported roles of SUR1-TRPM4 in different types of CNS injury, as well as the current progress and challenges in therapeutic inhibition.

## 2. SUR1-TRPM4

### 2.1. SUR1-TRPM4 Structure

SUR1, encoded by *ABCC8*, is a transmembrane protein in the ABC transporter superfamily [6]. Like other proteins in that family, SUR1 has a core structure comprised of two sets of transmembrane domains (TMD1 and TMD2), each containing six transmembrane α-helices, and two cytosolic nucleotide-binding domains (NBDs) [2,51,52,53] (Figure 1). The characteristic feature of SUR1 is an additional set of five N-terminal transmembrane helices (TMD0) which is connected to TMD1 via a long cytosolic CL3 linker loop. SUR1 contains two functionally distinct nucleotide binding sites (NBS): the degenerate NBS1, and the consensus NBS2—which has greater ATPase activity [54].

TRMP4 belongs to a superfamily of membrane proteins that includes 28 non-selective cation channels. Most proteins belonging to this group conduct both monovalent and divalent cations, including Ca^2+^ [55,56,57]. TRPM4 is Ca^2+^ activated, and is one of only two known ion channels in the mammalian genome that exclusively and non-selectively conducts monovalent cations [5,58]. TRPM4 is an independently functional ion channel, whereas SUR1 requires binding to a pore-forming subunit for functional capability [59,60].

SUR1 co-associates with TRPM4 (previously SUR1-regulated NC_Ca-ATP_ channels/SUR1-NC_Ca-ATP_) and forms a hetero-octameric structure comprised of four SUR1 subunits and four TRPM4 subunits (the pore-forming complex), as schematized in Figure 1 [5,11]. The association of SUR1 with TRPM4 doubles the affinity of TRPM4 for Ca^2+^-calmodulin and its sensitivity to intracellular Ca^2+^ [5].

SUR1 also co-associates with the pore-forming subunit, KIR6.2/*Kcnj11*, an ATP-sensitive potassium channel, to form K_ATP_ channels, whose role has historically been extensively studied in pancreatic β cells and diabetes mellitus [2,61,62]. Although both SUR1-Kir6.2 and SUR1-TRPM4 channels are both regulated by SUR1, these two associations have opposite functional effects in CNS injury: K_ATP_ channel opening leads to potassium efflux and hyperpolarization of cells [2,6,11,63,64], whereas SUR1-TRPM4 channel opening leads to influx of monovalent cations, resulting in depolarization of the cell, cytotoxic edema, blebbing, and oncotic cell death (Section 2.2) [6,7,8,11,65]. SUR1 can also associate with the inwardly rectifying potassium channel, Kir6.1/*Kcnj8* [66,67], but this combination has not been reported in nature.

### 2.2. SUR1-TRPM4—Discovery and Function

Alone, SUR proteins do not perform any recognized function; rather, they act as regulatory subunits for pore-forming subunits that together form multiple ion channels [2]. In 2001, Chen and Simard first reported a novel SUR1-regulated non-selective ATP and Ca^2+^-sensitive cation channel (subsequently identified as TRPM4) in adult rat reactive astrocytes isolated from the hypoxic inner zone of the gliotic capsule [6,7,8]. In these seminal patch-clamp experiments, channel opening and cell depolarization were mediated by nanomolar concentrations of intracellular Ca^2+^ and ATP depletion (Figure 2A) [7,8]. This resulted in cell swelling and blebbing (Figure 2B).

After CNS injury, SUR1 undergoes obligate association with the pore-forming cation subunit, TRPM4 [5,6]. SUR1-TRPM4 is not expressed in normal CNS tissue [6] but is upregulated de novo in different cells including astrocytes, neurons, endothelial cells, macrophages and microglia after forms of CNS injury including ischemic stroke [6,65,68], SAH [36,40], TBI [26], SCI [27,28], CNS metastases [41], and others [6,26,29,36,65,68,69,70]. SUR1-TRPM4 plays a crucial role in the formation of cerebral edema across the spectrum of cellular swelling, and what has traditionally been referred to as vasogenic edema [11,12,15,16,20,71]. Depleted ATP levels or nanomolar changes of Ca^2+^ concentration result in SUR1-TRPM4 channel opening, facilitating sodium entry into the cell, depolarization, and an oncotic gradient—ultimately producing cell swelling and accidental (non-programmed) cell death [65,72,73]. SUR1 contains high-affinity binding sites for sulfonylurea drugs and related compounds including glibenclamide (US adopted name, glyburide) and repaglinide, which bind with nanomolar or sub-nanomolar affinity and pharmacologically modulate SUR1-regulated channel activity [6]. This represents the basis for preclinical and clinical studies evaluating SUR1-TRPM4 inhibition in CNS injury, as discussed later.

Besides its role in channel formation/regulation, SUR1 also acts as a chaperone, and traffics functional channels to the plasma membrane [74,75,76]. The trafficking and cell surface expression of K_ATP_ channels are regulated by an endoplasmic reticulum (ER)-retention signal (RKR), present in both the SUR1 and Kir6.2 subunits. Association of SUR1 and Kir6.2 results in shielding of the ER-retention signal, permitting the trafficking of the channel complex to the cell surface [76].

### 2.3. SUR1-TRPM4: Biophysical and Pharmacological Properties

The original patch-clamp experiments that led to the discovery of SUR1-TRPM4 channels showed that the channel is permeable to all inorganic monovalent cations (Na^+^, K^+^, Cs^+^, Li^+^, and Rb^+^) with a single channel conductance of 25 to 35 picosiemens (pS), but impermeable to divalent cations including Ca^2+^ and Mg^2+^ [7]. Experiments using a series of organic monovalent cations with increasing cation size showed that the channel had a pore radius of approximately 4.1 nm. A nanomolar concentration of cytoplasmic Ca^2+^ induced channel opening, and was blocked by intracellular ATP (EC_50_, 0.79 μmol/L), but was unaffected by ADP or AMP.

SUR1-TRPM4 is blocked by first- and second-generation sulfonylureas, tolbutamide (EC_50_, 16.1 μmol/L at pH 7.4), and glibenclamide (EC_50_, 48 nmol/L at pH 7.4), respectively [6,8]. Channel inhibition by glibenclamide is the consequence of the prolonged and increased probability of long closed state of the channel, with no effect on open channel dwell times or channel conductance [8]. In vitro experiments revealed effective prevention of SUR1-TRPM4-regulated Na^+^ influx, cytotoxic edema, and oncotic cell death with glibenclamide administration [7]. In the presence of ATP, diazoxide and NN414 (SUR1 activators) increase SUR1-TRPM4 opening [6,8]. Sulfonylurea and related drugs access SUR1 binding sites via the lipid layer. The active un-ionized form of weak-acid drugs like tolbutamide, glibenclamide (pKa = 6.3), and repaglinide [77], result in increased drug binding [78], and more effective channel blockade for both K_ATP_ and SUR1-TRPM4 channels at acidic pH [77]. The magnitude of channel blockade in neurons from ischemic core brain tissue with 50 nM glibenclamide doubled when pH was decreased from 7.4 to 6.8 [65]. This pH-dependent drug potency enhances drug effects at the acidic pH characteristic of ischemic or injured CNS tissues [79]. In addition to its action on the closed/open state of SUR1-TRPM4, glibenclamide may also decrease surface expression of channel complexes; this has been noted for K_ATP_ channels, apparently due to abnormal trafficking induced by the drug [80].

### 2.4. SUR1-TRPM4: Transcriptional Regulation

Prior to the report of SUR1-TRPM4 in the CNS, SUR1 (encoded by *ABCC8*) had been extensively studied with regards to its role in regulating the K_ATP_ channel involved in insulin secretion in pancreatic β-cells. Earlier studies regarding transcriptional regulation of *Abcc8* were performed in models of diabetes [81,82]. In pancreatic β-cells, several transcription factors including Sp1, FoxA2/HNF3β, Beta2/NeuroD, and STAT3 have been reported to regulate the transcription of *Abcc8*; Sp1 in particular appears to play a critical role in the basal expression of the gene [81,82,83,84,85]. This section discusses key transcriptional upregulators of *Abcc8* and/or *Trpm4* that have been evaluated in CNS cells or models of CNS injury.

#### 2.4.1. Hif1a and Sp1

In the CNS, *Abcc8* upregulation reported during hypoxia/ischemia is regulated via a complex mechanism of sequential gene activation involving hypoxia-inducible factor 1α (*Hif1α*) [5,6,86]. In human brain microvascular endothelial cells, *Hif1α* expression during hypoxia increased luciferase reporter activity driven by the *Abcc8* promoter [86]. A series of luciferase reporter assays further revealed that activation of *Abcc8* transcription by *Hif1α* required binding sites for Sp1 (and not *Hif1*) [86]. Further examination of the Sp1 promoter using luciferase reporter assays and chromatin immunoprecipitation after cerebral ischemia revealed that *Hif* binds to *Hif*-binding sites on the rat Sp1 promoter to stimulate transcription of the *Sp1* gene [86]. Sp1, in turn, has been shown to induce *Abcc8* transcription in mice, humans, and rats [81,82,86]. These data, combined with earlier work in cerebral ischemia [65], suggest the crucial role of Sp1 in the regulation of *Abcc8* expression across species [81,82,86]. Pharmacological inhibition and gene suppression of *Hif* is protective in cerebral ischemia/hypoxia, which corresponds with the role of *Hif1* in upregulation of SUR1 in focal ischemia [87,88,89]. Similar results were found in the Rice–Vannucci model of neonatal hypoxia–ischemia, confirming neuroprotection [90,91].

#### 2.4.2. TNFa and NF-κB

TNFa and NF-κB are also important transcriptional regulators of *Abcc8*. The 5′ flanking region of both rat and human *Abcc8* promoter sites contains at least two consensus NF-κB binding sites [40]. In cultured brain endothelial (bEnd.3) cells, a 6 h exposure to TNFa (an activator of NF-κB), increased *Abcc8* mRNA and SUR1 protein levels [92]. Exposure of bEnd.3 cells to TNFa resulted in nuclear accumulation of the p65 NF-κB subunit. An electrophoretic mobility shift assay in bEnd.3 cells further demonstrated a physical interaction between the rat *Abcc8* promoter and NF-κB [40]. Additionally, the functionality of NF-κB binding sites in the *Abcc8* promotor was confirmed in studies using a luciferase reporter plasmid in the rat *Abcc8* promotor encompassing the recognized NF-κB binding site. TNFa stimulated luciferase activity ~2.3×, driven by NF-κB binding sites; rat *Abcc8* promotor activity also increased ~2× with TNFa [40]. Both Sp1 and NF-κB play an important role in transcriptional upregulation of *Abcc8* in trauma [30].

#### 2.4.3. Toll-like Receptor (TLR)-4

TLR4 has increasingly been recognized as initiating harmful neuro-inflammatory cascades, including via microglial activation [92,93,94]. In rats, TLR4-activated P2Y12^+^ microglia have shown markedly increased mRNA expression for both *Abcc8* and *Trpm4*, with similarly enhanced immunolabeling for the SUR1 and TRPM4 proteins [92]. These results were corroborated in vitro, and in addition to *Abcc8* and *Trpm4* mRNA upregulation (at 6 and 24 h after TLR4 ligation), *Il-1b* mRNA was also upregulated [92]. Changes in *Kcnj11* mRNA were not observed at either timepoint, suggesting that the K_ATP_ channel (SUR1-KIR6.2) is not involved in TLR4-activated microglia. Similar findings were also reported in N9 microglia where TLR4 activation resulted in a switch from a quiescent to an activated phenotype associated with upregulation *of Abcc8/Trpm4* mRNA and SUR1-TRPM4 channels [92].

### 2.5. SUR1 Pathways

The SUR1-TRPM4 channel interfaces with several other key secondary injury pathways [11], including those involved with cerebral edema (AQP4) [95], BBB integrity (matrix metalloproteinase-9 (MMP9), zona-occludens-1 (ZO-1)) [36,40,44,96], neuroinflammation (nitric oxide synthase-2 (NOS2), nuclear factor of activated T-cells (NFAT), calcineurin, calmodulin-dependent protein kinase II (CAMK-II) [36,40,45,92,97,98], and cell-death—including apoptotic pathways (Bcl-associated X protein (BAX), caspase-3; Table 1 and Figure 3).

#### 2.5.1. Cerebral Edema Pathways

AQP4 is a key mediator of CNS water influx and efflux, with a complex effect on cerebral edema, given its role in both edema formation and clearance [99,100,101,102,103,104]. AQP4 expression in astrocytes (especially in perivascular endfeet) has been shown to mediate water flux across the bloo–brain barrier—this increases in response to hypoxia-induced cell swelling, possibly in a calmodulin-dependent manner [105,106]. AQP4 channels on astrocyte vascular endfeet have recently been identified as central to the glymphatic system pathway of cerebral edema [102,103,104,107,108,109]. More comprehensive reviews focusing on AQP4’s role in cerebral edema have been reviewed elsewhere [105,107,108,109]. SUR1-TRPM4 has been shown to physically co-assemble with astrocytic AQP4 to form a novel hetero-multimeric water/ion channel complex [95]. In experiments using COS-7 cells, SUR1-TRPM4 co-immunoprecipitated with AQP4—both the M1 and M23 isoforms. FRET analyses showed that AQP4 directly co-associated with SUR1-TRPM4 to form a heteromeric channel complex [95]. Further in vitro studies in primary astrocytes demonstrated that close co-localization of TRPM4 and AQP4 leads to synergistically mediated, fast, high-capacity water influx resulting in astrocytic swelling [95]. This was validated in vivo, in a murine model of cerebellar cold injury [95]. Furthermore, it was noted that the co-assembly of SUR1-TRPM4 and AQP4 may amplify the role of TRPM4 as a negative regulator of Ca^2+^ influx, given the dilutional effects of water influx (in addition to the already increased TRPM4 Ca^2+^ sensitivity and reduced inward driving force/electrochemical gradient with SUR1-TRPM4 co-assembly) [95]. In a recent study of a rat model of SAH (Section 3.4.2), pituitary adenylate cyclase-activating polypeptide (PACAP38)-mediated downregulation of SUR1 also attenuated AQP4-mediated glymphatic dysfunction [110].

#### 2.5.2. BBB Permeability Pathways

MMP-9, tissue plasminogen activator (tPA) [6,96,111], and ZO-1 [44] have been implicated in BBB disruption and mechanistically overlap with SUR1-TRPM4 [11,44,96,111]. In NF-κB-activated brain endothelial cells, tPA induced SUR1-TRPM4 channel opening and MMP-9 secretion; both processes were dependent on protease-activated receptor 1 (PAR1) [96]. tPA converts plasminogen to plasmin, which in turn induces canonical activation of PAR1. Surprisingly, SUR1-TRPM4 inhibition (by glibenclamide and shRNA against *Abcc8*) reduced tPA-induced phasic MMP-9 secretion from activated endothelial cells. The precise relationship underlying this is unknown but it is speculated to be due to increased phosphorylated CamKII leading to desensitization and internalization of PAR1 [96].

The protective effect of glibenclamide in BBB permeability has also been studied in a mouse model of TBI [44]. TBI was induced in C57Bl/6 mice using controlled cortical impact (CCI). After controlled cortical impact (CCI) in mice, 10 μg of glibenclamide was given intra-peritoneally for 3 days. This showed a significant reduction in brain water content (*p* < 0.05), Evans blue extravasation (*p* < 0.01), and tissue hemoglobin levels (*p* < 0.05). Attenuation of apoptosis was observed with the glibenclamide treatment via the JNK/c-jun signaling pathway, ultimately resulting in elevated ZO-1 expression—thereby possibly preventing/minimizing BBB disruption.

#### 2.5.3. Neuroinflammation Pathways

The expression of pro-inflammatory cytokines (TNFα, IFNγ and IL-17) in models of SAH and experimental autoimmune encephalomyelitis (EAE) is decreased with both pharmacological inhibition of SUR1 or *Abcc8* gene suppression [36,97]. In EAE [98], global *Abcc8* deletion or pharmacological inhibition of SUR1 further reduced the inflammatory burden of several additional markers, as measured by the decreased expression of CD45, CD20, CD3, p65, TNFα, BAFF, CCL2, and NOS2.

In vivo and in vitro studies on microglia showed that activation of TLR4 by LPS led to de novo upregulation of SUR1-TRPM4 channels as well as CN/NFAT-dependent upregulation of *Nos2* mRNA and NOS2 protein [92]. In this study, it was shown that either gene silencing of *Abcc8* or *Trpm4* or pharmacological inhibition of SUR1 by glibenclamide, and of TRPM4 by 9-phenanthrol, resulted in downregulation of NOS2 expression.

In a rat model of blast traumatic brain injury, GFAP and Iba-1 were upregulated, indicating reactive astrogliosis and microglial activation [45]. In this study, pretreatment with glibenclamide led to downregulation of GFAP and Iba-1 expression.

#### 2.5.4. Cell Death Pathways

The role of SUR1 as it pertains to cell death pathways has predominantly been reported in SAH (caspase-3) and TBI (BAX-6/Bcl2). In a rat model of SAH, TNFα induced apoptosis through the activation of caspase 3—an effect that was reversed with glibenclamide, where treatment improved cell survival [40]. After murine CCI, the expression of the pro-apoptotic protein BAX was significantly upregulated, while that of the anti-apoptotic protein Bcl-2 was downregulated [44]. Glibenclamide treatment reversed these expression patterns.

## 3. SUR1-TRPM4 Expression and Inhibition in CNS injury

### 3.1. Ischemic Stroke

In the CNS, SUR1-TRPM4 (and its inhibition) has been most extensively studied in the context of ischemic injury, both in preclinical models as well as in human cohorts. These studies are summarized in Table 2 and Table 3.

#### 3.1.1. Ischemic Stroke—Preclinical Studies

##### Ischemic Stroke—In Vitro Studies

The original in vitro patch-clamp studies described earlier (Section 2.2 and Section 2.3) were performed on freshly isolated rat brain astrocytes after chemically induced hypoxia; these experiments demonstrated that depletion of cytosolic ATP resulted in activation of SUR1-TRPM4 channels and cell membrane depolarization [7]. The activation and opening of SUR1-TRPM4 resulted in an influx of Na^+^ and water into the cell, ultimately leading to osmotic gradient-driven cell swelling and blebbing [7,129]. These effects were reproduced in the absence of ATP depletion by treatment of reactive astrocytes with the SUR1 activator diazoxide, which resulted in channel activation; inhibitors such as glibenclamide and tolbutamide were protective [8].

##### Ischemic Stroke—In Vivo Expression Patterns

A subsequent seminal study by the same group was the first to demonstrate that this channel was upregulated in vivo in two rat models of ischemic stroke, and mediated cerebral edema in both models: massive middle-cerebral artery (MCA) infarction with malignant cerebral edema (MCE), as well as a non-lethal thromboembolic infarction [65]. In this work, SUR1 protein levels were increased in the infarct core within 2–3 h of MCA occlusion (MCAO) in “virtually every” neuron and capillary, and declined by ~8 h [65]. In the peri-infarct region, SUR1 levels increased later, but persisted for longer (8–16 h), with as much as a 3-fold elevation (vs. control) in several cell types including neurons, astrocytes, and endothelial cells. The elevated protein levels were corroborated by corresponding increases in *Abcc8* mRNA transcripts. A pilot porcine study after malignant MCA infarction also demonstrated elevated SUR1 and TRPM4 expression in the core (highest levels) and penumbra, as well as in contralateral neurons and microvessels 5 h after ischemia [119].

##### Ischemic Stroke—In Vivo Channel Blockade

In the MCE rat model, treatment with glibenclamide (75 ng/h) decreased mortality from 65% to 24%, with a significant reduction in peri-infarct edema (Figure 4). In the non-lethal thromboembolic model, glibenclamide reduced infarct volume at 2 days and 7 days [65]. This study was also the first to demonstrate preferential uptake of glibenclamide into ischemic tissue, as noted in Section 2.3—a finding that has since been confirmed [116]. Further evaluations of glibenclamide in different models of non-lethal stroke confirmed protection [112]. Delayed administration of glibenclamide, up to 6 h post-MCAO, still reduced total and/or corrected cortical lesion volumes by 41–53% measured at 48 h in thromboembolic MCAO, transient MCAO, and permanent MCAO models [112]. Doses used in these studies ranged from a loading dose of 3.3–10 mg/kg, and infusion doses of 75–200 ng/h.

The same investigative group further evaluated the in vivo benefits of glibenclamide in clinically relevant rodent models of stroke treatment, including decompressive craniectomy and recombinant tissue plasminogen activator (rtPA) [111,113,115]. In a model of severe ischemia/reperfusion, where reperfusion and glibenclamide treatment were initiated 6 h after the onset of ischemia, glibenclamide (10 mg/kg plus 200 ng/h) was as effective as decompressive craniectomy in reducing mortality, but was superior in preserving neurological function as well as watershed cortex and deep white matter [113]. Given the expression of SUR1-TRPM4 in the CNS microvascular endothelium, and the contribution to both cerebral edema and hemorrhage progression, glibenclamide was evaluated in a rat model of MCAO using an intra-arterial occluder followed by recanalization at 4.5 h and administration of rtPA (0.9 mg/kg × 30 min, identical to the human dose) [115]. Glibenclamide treatment at either 4.5 h or 10 h post-recanalization (10 mg/kg followed by 200 ng/h) improved hemispheric swelling, 48 h mortality (by >30%), and improved neurological outcomes as measured by Garcia scores. Interestingly, there was no difference in hemorrhagic transformation or 48 h infarct volume; the authors postulated that there was minimal opportunity for reducing infarct volume after a 4.5 h duration of ischemia—by which point the maximal ischemic insult had already occurred. The robust improvement in neurological outcomes with delayed glibenclamide treatment (as much as 10 h) further added to the growing body of evidence that glibenclamide is a promising therapeutic agent, and may provide benefits independently from the final infarct volume—possibly via effects on swelling or alternative neuroprotective pathways.

Independent groups have subsequently validated the protective effects of glibenclamide in preclinical models [116,117,118]. Wali et al. evaluated the same dose of glibenclamide as previously tested (10 mg/kg plus 200 ng/h infusion) in a permanent MCAO rat model [117]. Confirming the previous results, glibenclamide treatment significantly reduced infarct volume, hemispheric swelling, lowered neurological severity scores, and improved grip strength as compared to control rats [117]. Ortega et al. performed studies in transient MCAO rat models. They reported that treatment with different doses of glibenclamide (0.06 mg vs. 0.6 mg vs. 6 mg vs. vehicle) at 6 h, 12 h and 24 h, resulted in neuronal preservation within the infarct core (3 days after reperfusion), and improved neurological outcomes [116]. The dose of 0.6 mg had greater neuronal sparing vs. both 0.06 mg and vehicle, whereas 6 mg was only beneficial vs. vehicle but not the other doses. In this study, the beneficial effects of glibenclamide did not extend to the whole lesion volume (measured by MRI), nor did it protect against astroglial reaction. Immunolabeling studies identified a predominance of amoeboid reactive microglia within the necrotic core, co-expressing Kir6.2, SUR1 and SUR2B (at the time of this study’s publication, TRPM4 had not yet been identified as the NC_Ca-ATP_ inner pore, thereby precluding immunolabelling). This group also tested long-term effects of low dose glibenclamide (total = 0.6 mg) in rat transient MCAO [118]. Within 72 h of glibenclamide treatment, doublecortin-positive cells were noted to be migrating towards the ischemic cortex (increased vs. control); 30 days after MCAO, glibenclamide continued to enhance cortical neurogenesis, as evidenced by an increase in 5-bromo-2-deoxyuridine (BrdU) and NeuN labeling around the infarct. Given these histopathological findings, it is not surprising that glibenclamide was associated with behavioral improvement in both cognitive and sensorimotor functions up to 1 month [118].

Of note, in a rat model of ischemia/reperfusion (15 min/60 min), glibenclamide treatment reduced neutrophil recruitment, ameliorated inflammatory mediators (TNFa, prostaglandin E2), and boosted anti-inflammatory cytokine profiles [114]. However, involvement of SUR1-TRPM4 was not demonstrated in these studies and is thought to be unlikely, since the timepoints measured were insufficient for channel upregulation [6].

Glibenclamide has also been evaluated in combination with other therapeutic approaches in in vivo stroke models. In a rat MCAO model, glibenclamide and therapeutic hypothermia performed synergistically with multifactorial benefits [130]. Individual treatment with either glibenclamide or therapeutic hypothermia significantly reduced infarct volume. However, combination therapy was associated with greater edema reduction, less tight-junction loss, lower inflammatory cytokines (iNOS, COX2), and ultimately, higher performance on neurobehavioral tests vs. either therapy alone.

A recent report suggests that the antioxidant resveratrol may be a potentially novel therapy that targets SUR1 to provide benefit ischemic stroke; male rats were subjected to 2 h MCAO followed by intravenous resveratrol treatment (1.9 mg/kg) [120]. Resveratrol decreased *Abcc8* mRNA and SUR1 protein expression in MCAO and reperfusion, possibly by decreasing Sp1 binding activity. Resveratrol significantly reduced cerebral edema (water content), BBB disruption (Evans blue extravasation), and improved neurological outcome (limb-use asymmetry test) and survival (by ~40%). Resveratrol also reduced AQP4 expression in this study.

#### 3.1.2. Ischemic Stroke—Human Studies

##### Ischemic Stroke—Human Expression Patterns

The first systematic evaluation of SUR1 expression in brains from ischemic stroke patients was reported in 2013 [122]. In this study, postmortem tissue was obtained from 13 patients within the first 31 days after an ischemic stroke and evaluated for SUR1 expression using in situ hybridization and immunohistochemistry. SUR1 levels were elevated in all cases, with important temporal differences in cell-type-specific expression. Both neurons and endothelial cells were most elevated within the first week; expression was detected as early as 24 h after stroke onset, and persisted for 7–10 days. Conversely, both microglial and astrocytic cells showed a progressive increase in expression over the first month. Neutrophilic expression in peri-infarct regions remained prominent with no noticeable temporal changes. A subsequent study of post-mortem brain specimens from 15 patients (again, using both immunohistochemistry and in situ hybridization) confirmed TRPM4 elevation in all adult human brains [68]. TRPM4 upregulation persisted at 1 month post-infarction. FRET analysis confirmed the formation of SUR1-TRPM4 heteromers in neurons, endothelial cells, and astrocytes in human cerebral infarcts [68].

##### Ischemic Stroke—Clinical Retrospective Studies

Retrospective studies have evaluated the use of oral sulfonylureas (such as glibenclamide) in diabetic patients with ischemic stroke [123,124,125,126]. Given that SUR1-TRPM4 is upregulated after CNS injury, mechanistically it should be less likely that pre-treatment with glibenclamide would confer significant benefits. It is therefore not surprising that a Danish study of 4817 diabetics did not find an association between pre-admission sulfonylurea use and clinical outcome after ischemic stroke [126]. No pharmacokinetic values were available to evaluate the bioavailability of pre-treatment glibenclamide. In the VISTA (Virtual International Stroke Trials Archive), an analysis of 1050 patients identified 298 that were on pre-existing sulfonylurea medications [124]. Again, no association was noted between sulfonylureas and overall outcome, although there was a trend towards benefit in all 28 patients who were continued on sulfonylureas post-stroke. Of note, this cohort consisted of patients enrolled in non-reperfusion ischemic stroke trials, and stroke subtypes were not reported. In contrast, studies where sulfonylurea administration was continued post-stroke have indicated potential benefits. In acute ischemic stroke patients treated with sulfonylureas (from admission through discharge) 36.4% had a >4 point lower NIHSS score (National Institutes of Health Stroke Scale) at discharge vs. 7.1% in untreated patients (*p* = 0.007) ]. Treated patients were also more likely to attain an improved modified Rankin Scale score of ≤2 (81.8% vs. 57%, *p* = 0.035). A separate analysis of 220 diabetic patients with acute ischemic stroke reported that sulfonylureas reduced post-stroke symptomatic hemorrhagic transformation (0 vs. 11%, *p* = 0.016) and mortality (0% vs. 10%, *p* = 0.027) [125]. Brain tissue from one of the patients with symptomatic hemorrhagic transformation in this study demonstrated upregulated neuronal and microvascular SUR1.

##### Ischemic Stroke—Clinical Trials

RP-1127 (also known as Cirara) is an intravenous formulation of glyburide developed by Remedy Pharmaceuticals. After the RP-1127 program was purchased by Biogen, the drug was renamed BIIB093. This intravenous formulation was first evaluated in the field of ischemic stroke—specifically large hemispheric infarction, since this subtype is at highest risk for malignant cerebral edema.

A phase I trial (NCT01132703) in 34 healthy human volunteers [131] assessed the safety, tolerability and pharmacokinetics of RP-1127 at three doses: 3.0, 6.0, and 10.0 mg/day. No serious adverse events were reported. Although blood glucose levels ≥80 mg/dL were more common in the placebo group, there was no significant difference in levels ≤70 mg/dL with a dose of 3 mg/day, which was associated with steady-state drug levels of 27.3 ng/mL (higher than the 16 ng/mL required for efficacy in preclinical studies).

The safety profile of RP-1127 led to a phase IIa Glyburide Advantage in Malignant Edema and Stroke (GAMES-Pilot; NCT01268683) study. In this open-label, feasibility study with no control arm, a 72 h infusion of RP-1127 was tested at 3 mg/day in 10 patients with large hemispheric infarctions of 82–210 cm^3^ [132]. The drug was well tolerated without any safety concerns, symptomatic hypoglycemia, or adverse events requiring discontinuation or dose reduction. An exploratory analysis comparing these patients vs. historical untreated controls suggested improved clinical outcomes (*p* = 0.049) and a trend towards reduced mortality with treatment, although this was not statistically significant [49]. A case–control evaluation comparing the GAMES-Pilot patients with placebo-treated subjects in the Normobaric Oxygen Therapy in Acute Ischemic Stroke trial, revealed reduced vasogenic edema with RP-1127 treatment as measured by FLAIR ratio (*p* < 0.01), that persisted over time up to 80 h (*p* < 0.005) [25]. Compared with historical controls, RP-1127 also reduced MMP-9 levels at 48 h post-stroke onset as measured by quantitative sandwich ELISA (54 ± 17 ng/mL vs. 212 ± 151 ng/mL, *p* < 0.01) and pro-MMP-9 enzyme levels (as measured by zymography; *p* < 0.01) [25]. Both FLAIR ratio and MMP-9 levels are associated and known markers of BBB integrity [133].

The subsequent phase IIb study, GAMES-RP, was a randomized, multicenter, prospective, double-blind placebo-controlled trial that evaluated the same dose of RP-1127 as the GAMES-Pilot in 86 patients with large hemispheric infarction (lesion volumes 82–300 cm^3^) [47]. Treatment had to be initiated within ≤10 h of symptom onset, given the underlying biology of channel upregulation and preclinical data; 41 patients received RP-1127 and 36 received placebo. Overall infarct volumes (150–160 cc) were larger than in the GAMES-Pilot. The primary clinical outcome of mRS 0–4 without decompressive craniectomy was not different between groups, potentially due to widely variable surgical practice across centers. However, mortality at 30 days (regardless of craniectomy) was reduced in the treated group (15% vs. 36%, *p* = 0.03), with a trend towards reduction at 90 days (17% vs. 36%, *p* = 0.06). Measures of cerebral edema and BBB stability also showed a promising response to treatment: midline shift observed on MRI at 72–96 h was almost half in the treatment group vs. controls (4.6 mm vs. 8.5 mm, *p* = 0.0006; Figure 5A). Plasma MMP-9 levels also were lower in the treatment vs. placebo groups (211 ng/mL vs. 346 ng/mL; *p* = 0.006). No differences were noted in serious adverse events. A subsequent publication of pre-specified edema-adjudicated endpoints highlighted that treatment with RP-1127 resulted in a lower proportion of edema-related deaths (2.4% vs. 22.2%, *p* = 0.01) and other markers of clinical worsening, e.g., as measured by NIHSS increases of ≥4 (37% vs. 71%, *p* = 0.043) [134]. An exploratory post-hoc study of 65 patients from the GAMES-RP cohort who were ≤70 years of age identified lower mortality at all timepoints in RP-1127-treated patients (hazards ratio 0.34, *p* = 0.04). Although the trend towards benefit in functional outcome as measured by the modified Rankin Scale was not significant (*p* = 0.07), there were other significant treatment effects, including on the Barthel index (*p* = 0.03), a reduced plasma level of MMP-9 (189 vs. 376 mg/mL, *p* < 0.001), and decreased midline shift (4.7 vs. 9 mm; *p* < 0.001) [48]. A separate post-hoc exploratory analysis of the GAMES-RP cohort reported that treatment with RP-1127 reduced net water uptake (b = −2.80; *p* = 0.016); gray matter net water uptake contributed to a greater proportion of midline shift vs. white matter (*p* = 0.001).

These findings led to the ongoing a phase III clinical trial of RP-1127 or Cirara (now, BIIB093) in large hemispheric infarction, sponsored by Biogen. CHARM (Ciara in large Hemispheric infarction Analyzing modified Rankin and Mortality; NCT02864953), is a randomized, double-blind, placebo-controlled, parallel-group, multicenter international study designed to evaluate the efficacy and safety of BIIB093 for severe cerebral edema [46]. Inclusion requires infarctions measuring 80–300 cm^3^ on either MRI (diffusion weighted imaging), or CT perfusion, or an Alberta Stroke Program Early CT Score, (ASPECTS) of 1–5. Patients undergoing thrombectomy are included in this study, provided that the post-procedure infarct volume is within the required parameters. The planned primary outcome is the percentage of patients with improvements in 90-day functional outcome assessed via the modified Rankin scale score. The trial is actively enrolling; interim results are not yet available.

### 3.2. TBI

Scientific research into the role of SUR1-TRPM4 in secondary injury processes after TBI is rapidly evolving with insights from preclinical and clinical studies (Table 4) [11,24,26,29,30,31,32,33,34,35,38,42,43,44,135,136,137,138,139].

#### 3.2.1. TBI—Preclinical Studies

##### TBI—In Vivo Expression Patterns

Expression of SUR1 and/or TRPM4 (henceforth, SUR1 ± TRPM4) has been evaluated in rat models of TBI, with all but one study being performed in models of focal cortical impact [26,29,30,140]. In all reports, regardless of injury severity or specific model-type, SUR1 ± TRPM4 was consistently upregulated in multiple cell-types of the neurovascular unit. The spatio-temporal patterns described in the studies below were previously reported [11], and are re-summarized here (Figure 6).

The first report evaluating post-traumatic CNS SUR1 levels was in a rat weight-drop model of focal cortical contusion (10 gm dropped from 5 cm, velocity = 1 m/s) [26]. SUR1 was upregulated immediately beneath the impact as early as 3 h after injury. Upregulated expression was noted predominantly in microvessels and remained elevated up to 24 h after injury [26]. By the 24 h timepoint, neuronal and microvascular SUR1 levels were increased in several regions beyond the injured cortex, including the thalamus and hippocampus [26]. In a milder injury model (CCI, 10 gm dropped from 3 cm, velocity = 0.77 m/s), increased SUR1 protein and *Abcc8* mRNA expression was noted in hippocampal neurons, first at 6 h, and peaking at 12 h [30]. This study also demonstrated that increased Sp1 expression (a transcription factor involved in *Abcc8* expression) preceded that of SUR1. The same group has demonstrated important spatiotemporal differences in SUR1-TRPM4 expression in contusion core vs. penumbra after controlled cortical impact (4.5 mm tissue displacement, velocity = 1 m/s velocity, dwell time = 200 ms) [29]. In that work, expression of SUR1 in core tissue varied temporally; it was highest at 6 h, remained elevated at 12 h, decreased at 24 h, and was upregulated again at 72 h. Further evaluation of specific cell-types of SUR1 upregulation in the core was informative. During the first 24 h, SUR1 upregulation was predominantly in microvessels (with TRPM4 co-localization); however, at 72 h, SUR1 expression was in small round cells (likely microglia/macrophages) with co-expression of both TRPM4 as well as KIR6.2 subunits. Penumbral expression at all time points was in GFAP+ glia, consistent with astrocytes—again with all three subunits present (SUR1, TRPM4, KIR6.2).

##### TBI—In Vivo Channel Blockade

SUR1-TRPM4 inhibition, most often with glibenclamide, reduced cerebral edema, BBB permeability, hemorrhage progression, and functional deficits in multiple independent studies of preclinical TBI models [29,30,32,33,35,43,44,45,112]. Benefits have been reported across different species (mice, rats), severities, and models of TBI including fluid percussion injury (FPI), cranial blast injury, and controlled cortical impact (CCI), although most studies were performed in CCI. The maximal benefit is likely also in CCI/contusional injury; Glibenclamide was one of only two drugs demonstrating benefit in Operation Brain Trauma Therapy (OBTT). The OBTT consortium evaluates promising preclinical therapies in TBI in a multi-center, randomized, blinded fashion across three different models: FPI, penetrating ballistic-like brain injury, and CCI [141,142]. In OBTT, glibenclamide significantly improved lesion volume as well as motor function in CCI, without any benefits noted in FPI or penetrating injury [33].

In two independent laboratories, glibenclamide reduced BBB breakdown at 24 h by approximately 2-fold in both rat and mouse models of CCI [26,44]. A temporal analysis revealed a benefit of glibenclamide treatment on BBB integrity with reduced blood extravasation notable as early as 3 h; this became statistically significant by 6 h post-injury, and had maximal benefit by 12 h [26]. Much like the results obtained in preclinical models of SAH (Section 3.4) and ICH (Section 3.6), treatment with glibenclamide was associated with decreased loss of ZO-1 and occludin in endothelial cells [44]. Cerebral edema, as measured by wet/dry weight, has also been reduced after glibenclamide treatment in CCI, by as much as 66% [26,43,143]. Diffuse/contralateral edema that develops after CCI plus clinically relevant second insults such as hypotension (modeling hemorrhagic shock) has also been demonstrated to benefit from glibenclamide treatment [32].

Hemorrhage progression after CCI appeared to be halted/reduced by 45 min with no additional extravasated blood at 24 h compared with this hyperacute timepoint after glibenclamide treatment [26]. Similar results were observed with antisense oligodeoxynucleotides against *Abcc8* and *Trpm4*, with reduced hemispheric swelling and hemorrhage progression vs. controls after CCI [26,29]. Given the protective effects of glibenclamide on maintaining BBB integrity, it is not surprising that when hemorrhage progression after CCI was quantified, it was reduced by almost 60% in glibenclamide-treated rats [29]. Indeed, a third independent laboratory group demonstrated consistently reduced contusion volumes as measured by MRI at 8 h, 24 h, 72 h, and 7 days post-injury in a rat model of CCI [43]. In mice with global genetic deletion of *Abcc8*, contusion volumes were smaller, and hemispheres contralateral to injuries were larger (i.e., had less atrophy); this appeared to be influenced by sex, with less global atrophy in male but not female mice [35]. In vitro experiments suggest that these effects may be mediated by the SRY gene on the Y chromosome, and stimulation of *ABCC8* promoter activity [35]. This has not been reported with pharmacological channel blockade using glibenclamide, which may have different effects than constitutive, complete and global *Abcc8* knockout, including off-target actions that are not sex-dependent. Interestingly, in this study, *Abcc8* knockout naïve male mice had lower brain volumes, potentially suggesting a teleological role for SUR1 in CNS development that may be more important in males [35]. Further study of sex-based differences is warranted, particularly in the context of ongoing and future clinical trials (Section 3.2.2).

#### 3.2.2. TBI—Clinical Studies

##### TBI—Human Expression Patterns

Human expression patterns of SUR1 ± TRPM4 after TBI, particularly in contusional/pericontusional tissue, are analogous to patterns reported in rodent models (Section 3.2.1, Figure 6) [29,38,139]. In a cohort of 26 patients with contusions, SUR1 expression was markedly higher compared with controls (~3×, *p* < 0.001) [38]. Neuronal SUR1 was detected by 6 h post-TBI, peaking at 24 h. In contrast, levels remained persistently elevated up to 100 h in CD31^+^ endothelial cells. SUR1 levels were also moderately elevated in astrocytes (GFAP^+^), activated microglia, and neutrophils vs. control tissue. In a separate study of contusion specimens obtained from 32 patients, KIR6.2 was also found to be overexpressed in contusional astrocytes (consistent with findings in rodents) but not significantly elevated in either neurons or microglia vs. controls [139]. A nuanced map of regional SUR1, TRPM4 and KIR6.2 in the contusion core and penumbra was performed, analogous to the assessment in rodents by the same investigators (Section 3.2.1), and revealed a general consistency in results across species. SUR1-TRPM4 co-localization and co-assembly (by FRET imaging) was present in both microvessels and CD68^+^ round cells thought to be microglia/macrophages in the GFAP^−^ contusion core [29]. KIR6.2 was also expressed in the small round CD68^+^ cells but did not co-localize with SUR1. SUR1 expression in the GFAP^+^ penumbra was predominantly astrocytic rather than in microvessels/microglia, with heteromeric co-assembly of SUR1-TRPM4 as well as SUR1-KIR6.2. Neuronal expression of SUR1-TRPM4 in human contusion was not reported in this study.

SUR1 levels have also been explored in the CSF of adult and pediatric TBI patients [24,34]. The utility of SUR1 ± TRPM4 as a potential theragnostic biomarker is uniquely afforded by its de novo upregulation after CNS injury. Two small pilot studies have suggested markedly elevated CSF SUR1 levels in serial samples obtained from 28 adult [24] and 16 pediatric [34] patients with severe TBI. In adults, CSF SUR1 was elevated in all patients, vs. being undetectable in 15 controls with normal pressure hydrocephalus; in some patients, the ICP trajectory mirrored SUR1 expression after a temporal delay. Mean and peak SUR1 levels were associated with radiographic CT edema, and the initial degree of intracranial hypertension. Declining SUR1 levels between 48 and 72 h were associated with favorable prognosis: no patients had any intracranial hypertension or unfavorable outcomes [24]. In the pediatric cohort, samples (obtained from the “Cool Kids” randomized controlled trial), SUR1 expression was more nuanced: SUR1 was undetectable in all control CSF samples, and was elevated in 9 of the 16 patients with severe TBI. In this small sample, therapeutic hypothermia was not associated with SUR1 levels. CSF SUR1 was associated with increased ICP over 7 days (*p* = 0.004), and worse functional outcomes (*p* = 0.004). Patients who had SUR1 detected in any CSF sample after 24 h (regardless of level), had higher ICP values (*p* = 0.034) and worse functional outcomes at 12 months (*p* = 0.045) [34].

##### TBI—Human Genetic Variation

Based on the aforementioned biomarker studies, the same research group hypothesized potential genetic variation contributing to heterogeneity in SUR1 levels [135,136,137,144]. In a series of targeted investigations into the impact of *ABCC8* and *TRPM4* genetic variability on secondary injury after TBI in a single-center cohort of 385–485 patients with severe TBI, they identified several regionally clustered polymorphisms in both genes that were consistently associated with measures of intracranial hypertension, radiographic edema, and hemorrhage progression. The effect sizes were large, and findings were biologically consistent [135,136,137,144,145]. Although only approximately 1% of polymorphisms in *ABCC8* (and *TRPM4*) are linked with brain-specific mRNA levels (i.e., brain-specific expression quantitative trait loci; eQTL), all the single nucleotide polymorphisms identified as significantly associated with hemorrhage progression after TBI were eQTLs with biologically concordant effects. For example, *ABCC8* eQTLs associated with increased brain-specific *ABCC8* mRNA levels were also associated with increased odds of hemorrhage progression [145]. Regulatory annotations of these regions further revealed promotor and enhancer marks, as well as active brain-tissue transcription start sites. Regions in linkage disequilibrium with these spatially clustered polymorphisms encoded both the SUR1 site, as well as the interface/juxtaposition between the two subunits (i.e., SUR1 and TRPM4), suggesting potential functional or splicing consequences. Interestingly, significant interactions between *ABCC8* and *TRPM4* single nucleotide polymorphisms (SNPs) have also been reported, where certain genotype combinations containing risk alleles in both genes markedly and consistently increase the odds of several measures of intracranial hypertension [136]. The true consequences of these polymorphisms are yet to be evaluated in biological models, be it channel structure, function, regulation, expression, and/or post-translational modification. Nonetheless, identifying and validating patho-biologically relevant genetic variation in this pathway could have valuable implications for the design of future clinical trials, and could also inform novel targets and drug discovery (Figure 7). Given the small sample sizes available for genetic studies in TBI, unbiased genome-wide association studies have not yet validated these findings—but these important initiatives are ongoing in a large transatlantic multi-center effort known as GAIN (Genetic Associations in Neurotrauma). However, even such large-scale studies are limited in terms of severe TBI patients, and may be underpowered to detect contributions of *ABCC8-TRPM4* genetic variation to secondary injury.

##### TBI—Clinical Trials

Oral glibenclamide after TBI has been tested in two small randomized single-center clinical trials in Iran with encouraging results [42,138]. Randomization of 40 patients with diffuse axonal injury to 1.25 mg glibenclamide every 12 h for 1 week (or until ICU discharge) resulted in improved functional outcomes at discharge in the treatment group vs. control, as measured by the Glasgow Coma Scale (GCS) scores, and Glasgow Outcome Scale scores (all *p* < 0.004). In contusional TBI, randomization of 66 patients to either 10 mg oral glibenclamide vs. placebo resulted in lower contusion expansion ratios between baseline to day 3 (*p* < 0.001) and baseline to day 7 (*p* < 0.003). However, no differences were reported with absolute contusion volumes or functional outcomes [138].

Given the pharmacokinetic variability with intermittent oral glibenclamide dosing (supratherapeutic peaks, subtherapeutic troughs, influence of stomach pH on plasma levels), the intravenous formulation BIIB093 (see Section 3.1.2) was also tested in a small phase II multicenter randomized TBI trial [31]; 28 patients with GCS 4–14 were randomized to 72 h of BIIB093 within 10 h of injury; 14 of these patients had contusional TBI. Although the difference was not statistically significant (*p* = 0.15), the 7 patients with contusion who received the placebo had an increase in lesion volume by 1036% from baseline vs. the 7 with contusional TBI who received BIIB093, in whom lesion volume increased by 136%. The change in hemorrhage volumes was also not significant, likely due to the small sample size, but hemorrhage volumes in BIIB093 treated patients decreased 29.6% vs. increased by 11.6% with placebo. In MRI measures of edema, treated patients had no differences between lesional vs. uninjured white matter, unlike untreated patients, where increased edema was detected within the lesion (*p* < 0.02).

Driven by the promising preclinical and molecular data in contusional TBI, a larger Phase II trial (http://clinicaltrials.gov/: NCT03954041) entitled Antagonizing SUR1-TRPM4 To Reduce the progression of intracerebral hematomA and edema surrounding Lesions (ASTRAL) is actively recruiting patients with contusional TBI to specifically evaluate effects of BIIB093 in this injury subgroup [11]. This is a multicenter, double-blind, multi-dose, placebo-controlled, randomized, clinical trial sponsored by Biogen. Enrollment in ASTRAL of 160 patients across multiple centers in North America, Europe, and Japan is planned. Two doses will be tested (3 mg/day vs. 5 mg/day for 4 days) with treatment initiation within 6.5 h of injury. The tighter treatment window (vs. in large hemispheric infarction) is in the context of the underlying biology based on the temporal course of microvascular SUR1-TRPM4 expression, as well as the known risk-window for contusion expansion. The primary outcome of this trial is radiographic: contusion expansion by 96 h. Secondary outcomes include evaluating effects of BIIB093 on acute neurologic status, functional outcomes, and survival [11].

### 3.3. SCI

Traumatic spinal cord injury (SCI) is a devastating condition that results in blood vessel shearing and an immediate ‘primary hemorrhage’. Over time, similar to hemorrhage progression in TBI, this initial lesion may evolve/progress, resulting in a phenomenon termed “progressive hemorrhagic necrosis” (PHN) [6,9,146]. Given the expression of SUR1-TRPM4 in microvascular cells and their putative contributions to PHN, several studies have explored this channel and its inhibition in SCI with promising results [27,28,147,148,149,150], and a clinical trial is recruiting [151].

#### 3.3.1. SCI—Preclinical Studies

##### SCI—In Vivo Expression Patterns

In both mice and rat SCI models, SUR1 protein and *Abcc8* mRNA upregulation has been demonstrated in neurons, white matter, and microvascular cells using immunoblot analysis, immunohistochemistry, and in situ hybridization (Table 5) [27,28]. In severe unilateral SCI in rats (10 gm weight dropped from 2.5 cm), SUR1 upregulation was prominent in tissues surrounding the necrotic lesion; by 24 h, the lesion was larger, and SUR1 upregulation extended to distant tissues including the contralateral hemicord [28]. In the core lesion, labeling was noted in both neuronal and capillary-like structures, but in the penumbra, upregulation appeared limited to the microvasculature. SUR1 upregulation after SCI has also been demonstrated in mice, again in several cell types including oligodendrocytes as well as neurons [27]. Similar expression patterns have been observed in mice, rats, and humans, indicating cross-species preservation of this pathway after spinal trauma [27]. Concurrent neuronal and microvascular upregulation of TRPM4 in SCI rodent models has also been demonstrated [5,28,152]. Co-immunoprecipitation experiments and FRET analyses have demonstrated the formation of abundant SUR1-TRPM4 heteromers in multiple cell-types, including those of the microvasculature [5]. In all the aforementioned studies, uninjured spinal cords had minimal labelling.

##### SCI—In Vivo Channel Blockade

The benefits of channel blockade in SCI have been demonstrated using several strategies including gene deletion, suppression (anti-sense oligodeoxynucleotides, ODN), and pharmacological inhibition of either the SUR1 or TRPM4 subunit [27,28,147,148,149,150,152]. All approaches have yielded similar histological and functional results.

Unilateral SCI in both *Abcc8*^−/−^ mice and *Trpm4*^−/−^ mice reduced secondary hemorrhage and lesion expansion, prevented PHN, resulted in minimal capillary fragmentation, and improved functional outcomes at 1 week vs. wild-type [27,152]. These findings were corroborated in rats treated with anti-sense ODNs against *Abcc8*; lesion volumes were reduced to 25% of controls [27]. Pharmacological inhibition of SUR1-TRPM4 has been reported using SUR1-binding drugs (glibenclamide and repaglinide) or via targeting sodium currents and TRPM4 (riluzole, flufenamic acid), as shown in Table 5 [5,27,28,147,148,149,150,152,154,155]. However, specific pharmacological blockade of TRPM4 has not been reported, likely due to the non-specificity of drugs for this subunit in the context of structural similarities with other ion channels [9]. Riluzole, for example, also inhibits excitotoxicity—potentially via blocking glutamate release.

To date, seven studies of pharmacological SUR1-TRPM4 inhibition in SCI have been reported, six from the same group, with confirmatory findings from an independent laboratory—all of which have demonstrated benefits regardless of the laterality and initial severity of injury. One of the first reports was in a rat model of unilateral cervical spinal cord injury: here, PHN was associated with upregulation of SUR1 in a time-dependent manner in capillaries and post-capillary venules [28]. Glibenclamide reduced PHN and capillary fragmentation within 24 h, spared both contralateral and ipsilateral white matter tracts, and improved both lesion volume and neurobehavioral assessments at 7 days [28]. Lesion volumes were reduced approximately 3-fold compared to controls. In this study, the reduction in PHN was reproduced with in vivo gene suppression of *Abcc8* using antisense ODNs, which are taken up preferentially by penumbral microvessels in the spinal cord [28]. Similar results were obtained independently by another group, confirming the attenuation of PHN and improvement in functional outcomes after SCI with glibenclamide treatment [150]. The benefits of glibenclamide extended to bilateral primary SCI in a rat model: glibenclamide-treated rats had improved functional benefits as early as 24 h, and reduced lesion volumes at 6 weeks. Nonetheless, the effect sizes were smaller compared to unilateral SCI. Lesion volumes after treatment with glibenclamide in bilateral SCI were 33% smaller than controls, vs. 57% reduction after unilateral injury [148].

The benefits of glibenclamide after unilateral rat cervical SCI are measurable radiographically. T2 sequences on MRI proved to be an accurate, non-invasive imaging biomarker in a study of 28 female rats [154]. Here, an approximately 2.3-fold expansion in lesion volume was quantified within 24 h after injury in control rats, but only a 20% increase in those treated with glibenclamide. These values corresponded closely with measures of hemorrhagic contusion in tissue sections. This has important implications for future clinical trial design, including in terms of quantifying and endophenotyping the treatment effects of SUR1-TRPM4 inhibition.

Riluzole has been reported to significantly improve motor recovery, locomotion and functional neurological outcomes in a variety of SCI animal models [156]. Comparisons between glibenclamide (200 ng/h) vs. riluzole in a rat model of cervical SCI associated with high mortality, revealed similar acute improvement after either drug in terms of capillary fragmentation, PHN, Basso, Beattie and Bresnahan locomotor scores, and mortality [149]. However, glibenclamide resulted in better tissue sparing, as well as improved measures of complex function (grip strength, rearing, accelerating rotarod) [149]. Of note, both drugs were initiated at 3 h post-injury, and continued for 7 days. A subsequent study compared glibenclamide, riluzole, and systemic hypothermia after unilateral rat SCI, with further delayed administration of treatment at 4 h post-injury, as well as a higher dose of glibenclamide (400 ng/h) [147]. In this work, mortality was high without treatment (30%), and also with riluzole (30%). Reduced mortality was noted with hypothermia (12.5%), and none of the rats in the glibenclamide-treated group died. Amongst the survivors, glibenclamide and hypothermia demonstrated similar efficacy, as measured by locomotor scores (modified Basso, Beattie and Bresnahan); riluzole was inferior to both treatments in the acute setting (2 weeks). However, all three treatments were similar by the last four weeks of the study. Lesion volumes vs. controls were reduced in all treatment groups, with the maximal benefit seen in glibenclamide-treated rats.

Drugs other than glibenclamide that impact MMP-9, SUR1 and/or TRPM4 have also been evaluated after SCI [153,157,158]. In a rat model of moderate thoracic SCI, immediate intraperitoneal treatment with ghrelin (80 mg/kg, followed by the same dose every 6 h for 1 day) decreased blood–spinal cord barrier (BSCB) dysfunction, reduced MMP-9, SUR1 and TRPM4 expression, and lessened macrophage and neutrophil infiltration after injury [158]. These effects were reported to be mediated by ghrelin receptor 1A; the benefit was lost when rats were administered a receptor antagonist. In a separate study using the same model of SCI, 150 mg/kg of immediate mithramycin A (MA) injection (followed by daily injections of the same dose for 5 days) decreased blood–spinal cord barrier breakdown, inhibited infiltration of neutrophils and macrophages, and decreased the expression of MMP-9, SUR1 and TRPM4 [157]. In this study, MA treatment reduced apoptosis and improved functional recovery. Recently, flufenamic acid (FFA) was evaluated in a thoracic SCI model using moderate contusion at T10; intraperitoneal FFA was injected 1 h after injury, and then dosed daily for 1 week [153]. Treatment with FFA inhibited TRPM4 expression (immunohistochemical localization, RT-PCR, western blot), secondary hemorrhage, and capillary fragmentation, and promoted angiogenesis (immunohistochemistry for vWF and KI-67 co-labeled vessel) [153]. FFA significantly downregulated the expression of MMP-2 and MMP-9 at 24 h after SCI, and significantly attenuated blood–spinal cord barrier (BSCB) disruption at 1 day and 3 days after injury. FFA treatment protected motor neurons and improved locomotor function following SCI.

Similar to TBI, the SUR1-TRPM4 pathway and its inhibition in traumatic SCI may have differential effects based on sex—in this case, hormonally mediated [159]. In a rat model of moderate thoracic SCI, immediate treatment of male rats with 17b-estradiol (E2, 300 mg/kg, followed by the same dose at 6 h and 24 h post-injury) reduced BSCB breakdown, progressive hemorrhage, and infiltration of inflammatory cells such as neutrophils and macrophages. SUR1, TRPM4, MMP-9 and ZO-1 expression were also all reduced with E2 treatment. Functional improvement was also noted. All protective effects were negated with treatment by an estrogen receptor antagonist [159].

#### 3.3.2. SCI—Clinical Studies

##### SCI—Human Expression Patterns

Autopsy samples from seven patients who died from traumatic SCI demonstrated prominent penumbral SUR1 expression that tapered distally from the epicenter [27]. Upregulation was noted for both SUR1 protein (immunofluorescence) and *Abcc8* mRNA (in situ hybridization) expression. Similar to findings in rodent models (Section 3.3.1), expression in humans was demonstrated in both white matter, microvascular endothelial cells, and neurons.

##### SCI—Clinical Trials

To date, no results from clinical trials testing glibenclamide or other pharmacological SUR1-TRPM4 inhibition in SCI have been reported. A pilot open label multicenter prospective evaluation of oral glyburide entitled SCING (Spinal Cord Injury Neuroprotection with Glyburide, NCT02524379) is active but not currently recruiting. The primary objective of this initial phase multi-center open-label pilot study is to enroll 10 patients with traumatic SCI to assess the feasibility and safety of receiving oral glyburide within 8 h of injury [151]. Secondary objectives include both radiographic and blood-based biomarker discovery, as well as impact on neurological recovery.

### 3.4. SAH

Secondary brain injury after aneurysmal SAH involves cerebral edema, vasospasm, neuroinflammation, delayed cerebral ischemia and both early and delayed impairment in cognition. Emerging data suggest the involvement of the SUR1-TRPM4 channel in these processes after SAH (Table 6) [36,37,40,128].

#### 3.4.1. SAH—Preclinical Studies

##### SAH—In Vivo Expression Patterns

In a rat model of mild–moderate SAH using a filament-based endovascular puncture of the internal carotid artery, in situ hybridization detected strong expression of *Abcc8* mRNA in neurons and microvessels in the inferomedial cortex (i.e., adjacent to the region of SAH) [40]. Immunoblot and immunohistochemistry demonstrated SUR1 protein upregulation at 24 h—particularly in neurons and microvascular cells. This was noted both in the path of direct injury (inferomedial cortex) as well as in distal structures in the posterior cerebral artery territory. In vitro studies in the same report demonstrated that SUR1 transcription was activated by TNFa. Subsequent work from the same laboratory studied SUR1-TRPM4 expression 24–48 h after SAH. Both subunits were upregulated in cortical microvessels, neurons and astrocytes, with minimal expression of KIR6.2 [160]. Co-association of SUR1-TRPM4, but not KIR6.2, was confirmed using FRET analysis. Compared with controls, co-immunoprecipitation experiments revealed an 8-fold increase in co-associated SUR1-TRPM4.

##### SAH—In Vivo Channel Blockade

Both studies presented in Section 3.4.1 also evaluated SUR1-TRPM4 channel inhibition with glibenclamide after SAH. Treatment with a 10 μg/kg loading dose followed by 200 ng/h reduced BBB disruption, as measured by IgG extravasation, and decreased both local inflammation (TNFa and NF-κB immunolabeling) as well as reactive astrogliosis [40]. Moreover, in this study, rats treated with glibenclamide had minimal or no caspase-3 activation in the posterior cerebral artery territory, unlike untreated rats, suggesting that this drug may reduce TNFa induced apoptosis.

Two different rat models of SAH demonstrated similar benefits with SUR1 inhibition: the first with blood injection into the subarachnoid space of the entorhinal cortex, and the second with filament puncture of the internal carotid artery [36]. In both models, antisense ODN against *Abcc8* and glibenclamide reduced SAH-induced BBB disruption. These interventions also reduced TNFa expression in adjacent cortex by approximately 4-fold. The benefits of glibenclamide extended to tissue preservation as quantified by Black Gold II, Fluoro-Jade C, and DAPI staining of pyknotic nuclei. In models with entorhinal SAH, glibenclamide-treated rats had improved functional outcomes, including platform search strategies and rapid spatial learning tasks.

The neuropeptide PACAP (pituitary adenylate cyclase-activating polypeptide) has recently demonstrated potential benefits in rat SAH via a mechanism that possibly involves SUR1 [110]. Exogenous PACAP38 treatment preserved BBB function, accelerated CSF movement/clearance, attenuated 24 h brain edema, and improved neurological functional deficits, as measured by modified Garcia scores and beam balance tests [110]. In this study, the PACAP receptor (PAC1) signaling pathway involving adenylate cyclase, cyclic AMP, and protein kinase A was thought to contribute to neuroprotection via phosphorylation and degradation of SUR1. Increased SUR1 expression after SAH was associated with MMP-9 elevation and ZO-1 reduction; PACAP38 reversed these effects—MMP-9, SUR1, and AQP4 expression were reduced, and ZO-1 expression was preserved [110].

#### 3.4.2. SAH—Clinical Studies

##### SAH—Human Expression Patterns

Human SUR1 expression has been evaluated by immunohistochemistry in autopsy specimens from seven patients with aneurysmal SAH and compared with five normal brains with documented absence of SAH, ischemia or identifiable CNS pathology [36]. In all cases of SAH, SUR1-TRPM4 was abundantly expressed in microvessels, neurons, and astrocytes; based on semi-quantitative analysis, expression was always greater than levels noted in control brains, although the relative abundance varied between cases. FRET analysis demonstrated formation of SUR1-TRPM4 heteromers in both neurons and microvessels. A prospective study of 44 consecutive patients with aneurysmal SAH reported elevated serum SUR1 and TRPM4 levels (on days 1, 4, and 14 post-bleed) vs. controls—the latter recruited from outpatient headache clinics with negative head imaging on CT and/or MRI [37]. In this pilot study, day 14 serum SUR1 (*p* = 0.001) and TRPM4 (*p* = 0.044) levels were correlated with the GOS score.

##### SAH—Clinical Trials

There are no reports of SUR1-TRPM4 inhibition in human SAH. A randomized double-blind clinical trial, GASH (Glibenclamide in Aneurysmatic Subarachnoid Hemorrhage), is currently recruiting in São Paulo (NCT03569540). The investigators are evaluating the benefits of 5 mg of oral glibenclamide for 21 days after aneurysmal SAH, with the goal of enrolling 50 patients [128].

### 3.5. Cardiac Arrest

Cerebral edema and neurological devastation are hallmarks of cardiac arrest, with no currently available pharmacological agents that proffer benefit. Varying degrees of hypothermia or targeted temperature management have shown promising results in clinical trials, but the data are nuanced and beyond the scope of this review [161,162,163,164,165,166,167]. The role of SUR1-TRPM4 in secondary injury after global ischemia from cardiac arrest has been reported, but is less well developed compared with focal ischemia from vessel occlusion.

#### 3.5.1. Cardiac Arrest—Preclinical Studies

##### Cardiac Arrest—In Vivo Expression Patterns

In a 7 min murine model of asystolic cardiac arrest from KCl injection, *Abcc8* and *Trpm4* mRNA levels were significantly upregulated at 6 h post-cardiopulmonary resuscitation; *Trpm4* levels remained elevated at 24 h [168]. In this model, hypothermia reduced *Abcc8* and *Trpm4* levels. *Abcc8* and *Trpm4* upregulation was accompanied by TNFa, IL-6, and NF-κB upregulation. Protein expression was not evaluated in this study. These findings were similar in rat models of asphyxial cardiac arrest. After 8 min asphyxial cardiac arrest, both *Abcc8* and *Trpm4* mRNA, as well as SUR1-TRPM4 protein (western blot), were upregulated in the cortex and hippocampus at 24 h after injury [169]. In a 10 min rat model of asphyxial cardiac arrest, both *Abcc8* and *Trpm4* mRNA, as well as SUR1-TRPM4 protein (by western blot and immunohistochemistry), were upregulated at 6 h and peaked at 24 h [170]. Immunohistochemistry demonstrated that positive cells were predominantly in the cortex and CA1 hippocampal region; by 72 h, SUR1 was expressed in several cell types including neurons, endothelial cells, microglia, and astrocytes.

##### Cardiac Arrest—In Vivo Channel Blockade

One of the first studies of glibenclamide in cardiac arrest was performed in a rat model of 8 min asphyxial cardiac arrest [169]. Glibenclamide treatment (10 mg/kg loading dose, 1.2 mg at 6, 12, 18 and 24 h) increased 7-day survival, reduced neurological deficit scores, and reduced hippocampal neuronal loss vs. the untreated vehicle group. This study also demonstrated histopathological evidence of neuroprotection (reduced apoptosis and neuronal necrosis) acutely, at 24 h. Furthermore, glibenclamide reduced TNFa and monocyte chemokine protein 1 levels after return of spontaneous circulation. The same group compared the effects of glibenclamide with targeted temperature management after rat asphyxial cardiac arrest, and found similar levels of protection between the two treatment strategies in terms of reduced neuronal injury and neurological deficit compared to untreated rats at several time points up to 7 days [170]. A follow-up study by the same investigators utilized MRI parameters for quantification of benefits [171]. Here, cardiac arrest caused significant abnormalities with diffusion restriction, demonstrated by hyperintense diffusion weighted imaging at 72 h. This corresponded with histopathological evidence of neuronal swelling, dendritic injury, and microglial/astrocytic activation. The abnormal diffusion restriction was alleviated by glibenclamide treatment within 72 h. Glibenclamide also had a trend towards reducing neuronal loss at this time point, and significantly improved neurological deficit using the neurodeficit score.

An independent group evaluated the effects of glibenclamide after cardiac arrest in mice using a KCl injection model and evaluated the impact of the drug vs. the effects of hypothermia [168]. Similar findings were reported; glibenclamide significantly reduced brain water content at 24 h and improved BBB integrity with effects similar to those observed with hypothermia to 33 °C.

#### 3.5.2. Cardiac Arrest—Clinical Studies

##### Cardiac Arrest—Human Expression Patterns

There are currently no studies reported of SUR1-TRPM4 expression after cardiac arrest in humans.

##### Cardiac Arrest—Clinical Trials

There are currently no reported or ongoing clinical trials of SUR1-TRPM4 inhibition after cardiac arrest.

### 3.6. ICH

The evaluation of SUR1 ± TRPM4 and pharmacological inhibition of the channel by glibenclamide in primary ICH is less mature than current data in disease subtypes of ischemic stroke and TBI. Existing studies in ICH have yielded mixed results (Table 7) [143,160,172,173,174,175,176,177].

#### 3.6.1. ICH—Preclinical Studies

##### ICH—In Vivo Expression Patterns

Aside from subarachnoid hemorrhage, two of the earliest studies evaluating this pathway in primary hemorrhagic disease were in hemorrhagic encephalopathy of prematurity/germinal matrix hemorrhage [160,177]. The first, studied in human infant tissue, is summarized below (Section 3.6.2). In the rat model of hemorrhagic encephalopathy of prematurity, 20 min of intrauterine ischemia followed by an intraperitoneal injection of glycerol (to elevate venous pressure) resulted in periventricular hemorrhages in rat pups [160]. Similar to the findings in human premature infants, SUR1 protein was upregulated in rat periventricular tissue by 24 h after intrauterine ischemia, and was negligible in control tissues. SUR1 upregulation was noted in several regions including the choroid plexus, ependymal lining of the lateral ventricles, internal capsule, subventricular zone, corpus callosum and hippocampus, but most prominently in microvascular cells. In an autologous blood-induced rat model of ICH, SUR1 but not KIR6.2 protein was found to be upregulated by immunofluorescence in perihematomal neurons and endothelial cells at 24 h [172]. No upregulation was noted in microglia at either 24 h or 72 h after ICH.

##### ICH—In Vivo Channel Blockade

In the rat model of hemorrhagic encephalopathy of prematurity described above, a low non-hypoglycemic dose of glibenclamide (10 mg/kg load plus 400 ng/h infusion × 1 week) administered to the mother at the end of pregnancy protected the rat pups from insults of interuterine ischemia and postnatal glycerol-induced elevated venous pressure [160]. The number and extent/severity of brain hemorrhages were acutely reduced, and protection persisted into development; treatment of the mother decreased developmental delay in the pups and preserved brain and body mass.

Studies in adult models of primary ICH have yielded conflicting results. In the autologous blood-induced rat model of ICH where SUR1 upregulation was reported (Section 3.6.1), investigators reported that treatment with low-dose glibenclamide (10 μg/kg load plus 200 ng/h infusion for an unreported length of time) resulted in marked benefits, both histopathologically as well as functionally [172]. Brain water content and MMP expression were reduced, and BBB integrity was restored (with less Evans Blue extravasation, less attenuation of ZO-1, vWF, and occludin vs. vehicle) with glibenclamide treatment. Additionally, treated mice showed improved performance in the Morris water maze [172]. These benefits on functional outcome, brain edema, and BBB integrity were independently corroborated by two groups: one in a model of autologous blood infusion [143], and the other in a collagenase-induced ICH model [175]. In the collagenase-induced ICH model, glibenclamide regulated iNOS levels in microglia, increased the BCL-2/BAX ratio, and also reduced perihematomal caspase-3 expression and apoptosis [175]. In the autologous blood infusion mouse model, glibenclamide appeared to act via the NLRP3 inflammasome, even at a single dose of 10 mg/mouse [143]. In this study, treatment reduced perihematomal levels of several inflammatory cytokines (IL-1b, IL-18, IL-6, TNFa) vs. controls, and reduced the number of apoptotic cells (particularly microvascular/endothelial cells). Genetic deletion of NLRP3 eliminated the protective effects of glibenclamide on the BBB, suggesting that the NLRP3 inflammasome is involved in this pathway, and in mediating at least part of glibenclamide’s benefit.

Two recent preclinical reports have suggested no benefit from glibenclamide treatment after ICH [173,174]. Both studies were performed in rats with collagenase-induced ICH; one involved severe ICH [173], and the other was more modest ICH without mortality [174]. In both reports, glibenclamide was well tolerated, but did not improve edema measures, hematoma volume, or functional outcomes at any timepoint. In the severe ICH model, glibenclamide may have demonstrated increased cytotoxic edema, with larger cell body volumes noted in the perihematomal at 24 h (*p* = 0.001).

It is challenging to reconcile the discrepant results between these two studies and the previous body of work, particularly since both the species (rat) and the drug dosing (10 mg/kg, 200 ng/h continuous infusion) appear to have been consistent across studies. One possibility is the timing of administration: in the studies demonstrating benefit [143,172,175], glibenclamide was loaded either 30 min before or immediately after injury, whereas in those where no benefit was observed, drug was initiated 2 h after injury [173,174]. However, for ultra-early administration to be clinically translatable, it would require drug infusion in a prehospital setting. It is also possible that the collagenase-induced ICH models did not activate pathways that upregulate/trigger SUR1-TRPM4. Indeed, in the severe ICH model, there was no increase in *Abcc8* or *Trpm4* mRNA in the perihematomal region (protein expression was not evaluated) [173]. Another possibility is that, particularly in severe ICH, the injury is too overwhelming for any drug to demonstrate functional benefit, particularly if administration is delayed. Regardless, these studies cumulatively suggest that further preclinical investigation of SUR1-TRPM4 and glibenclamide is warranted in ICH.

#### 3.6.2. ICH—Clinical Studies

##### ICH—Human Expression Patterns

In brain specimens obtained from 12 premature infants (7 with histopathological evidence of germinal matrix hemorrhage), the most prominent expression of HIF1α was noted in infants with frank germinal matrix hemorrhage [177]. SUR1 expression was strongly present in nearly all veins in specimens from infants with frank germinal matrix hemorrhage, but was minimal in veins from infants without this injury [177].

##### ICH—Clinical Trials

A randomized clinical trial across five centers in China evaluating the impact of low-dose oral glibenclamide has recently been completed (NCT03741530) [127]. The goal of this study entitled “Glibenclamide Advantage in Treating Edema After Intracerebral Hemorrhage (GATE-ICH) was to enroll 220 patients with primary ICH, and evaluate both safety and efficacy. Outcome measures included functional scores at day 3, 7, and 90, as well as secondary assessments of midline shift, change in hematoma volume, perihematomal edema, and mortality. Results from this study have not yet been reported.

### 3.7. Multiple Sclerosis (MS) and Experimental Autoimmune Encephalitis (EAE)

MS is an autoimmune disease characterized by chronic inflammation, demyelination, and neurodegeneration of the CNS, which leads to a neurological deficit in young and middle-aged adults [179,180]. There is no known cure for MS. The principal animal model for MS, EAE, is characterized by myelin-specific autoreactive T cells that enter the CNS and initiate inflammation and tissue damage, resulting in oligodendrocyte cell death, axonal demyelination, and neuronal degeneration [97,181]. In both MS and EAE, inflammation is perpetuated by both infiltrating immune cells and astrocytes (Table 7) [182,183,184,185]. The role of SUR1-TRPM4 in the inflammatory pathway as it pertains to models of MS/EAE is still being explored, but early work suggests its involvement.

#### 3.7.1. MS and EAE—Preclinical Studies

##### EAE—In Vivo Expression Patterns

Upregulation of both SUR1 and TRPM4 have been reported in murine EAE induced by myelin oligodendrocyte glycoprotein 35–55 (MOG_35–55_) [97,178]. The earlier report focused on the TRPM4 subunit in both mice and humans (Section 3.7.2); here, by 14 days after immunization, murine TRPM4 expression was increased within axonal processes at the edge of EAE lesions (compared with healthy wildtype mice, *p* = 4.15 × 10^−13^) [178]. Overall, TRPM4+ cells were noted to be irregularly shaped and fragmented. Minimal changes in TRPM4 expression were detected in neuronal somata. Expression of SUR1-TRPM4 heteromers in EAE mice was subsequently reported by an independent group [97]. Here, by post-induction day 10, modest immunolabelling of SUR1 was noted in the white matter of mice with EAE that was widespread by post-induction day 30; quantitative analyses revealed a progressive increase with time. Expression was predominantly seen in astrocytes, with minimal co-localization in microglia and oligodendrocytes. Consistent with the prior study, most neuronal cells showed minimal SUR1. SUR1 co-localized with TRPM4; the formation of heteromers was confirmed with immunoFRET and co-immunoprecipitation experiments. The channel was not detected in normal wildtype controls.

##### EAE—In Vivo Channel Blockade

The benefits of SUR1 ± TRPM4 blockade in EAE have been demonstrated with genetic silencing as well as pharmacological inhibition using glibenclamide [97,98,178]. Compared with wild type untreated mice, in the MOG_35–55_ EAE model, both *Abcc8^−/−^* and glibenclamide treatment (on day 10) consistently resulted in improved clinical scores with robust effect sizes (Cohen’s *d* = 1.2) [97]. Treatment effects were similar between glibenclamide and *Abcc8^−/−^*, and persisted from post-induction days 10–30. No hypoglycemia was present in glibenclamide-treated mice. At post-induction day 30, sections from the lumbar spinal cord in *Abcc8^−/−^* mice were “indistinguishable” from normal wildtype controls, and had no evidence of constitutive inflammation [97]. The spinal cords of glibenclamide-treated and *Abcc8^−/−^* mice with EAE had reduced meningeal, perivascular, and parenchymal inflammatory infiltrates vs. untreated EAE mice, including cells expressing CD45, CD3 (T cells), CD20 (B cells), and CD11b (macrophages/microglia), as well as a reduced number of cells positive for pro-inflammatory cytokines such as TNFa, IL-17, and IFN-g [97]. Both genetic and pharmacological inhibition demonstrated improved myelin preservation, more mature and precursor oligodendrocytes, and protection against axonal damage in the EAE model. Subsequent work by the same group demonstrated that even later glibenclamide treatment, initiated during the chronic phase of EAE on post-induction day 24, demonstrated benefits, including reducing clinical severity, as well as decreasing the histopathological burden of demyelination and inflammation as measured by the same markers [98]. Mechanistically, in vivo effects of glibenclamide and global *Abcc8* deletion both appeared linked to reduced expression of TNF, BAFF (B cell activating factor), CCL2 (chemokine C-C motif ligand 2) and NOS2 (nitric oxide synthase 2) vs. untreated mice [98]. Interestingly, although both axonal and neuronal injury were reduced and mean clinical scores were improved in Trpm4^−/−^ mice subjected to the MOG_35–55_ EAE by an independent group, the autoimmune response was not modified by genetic Trpm4 deletion [178]. Much like the similar effects of *Abcc8*^−/−^ and glibenclamide in the EAE model, this study also reported almost identical effects of Trpm4 genetic deletion and glibenclamide in terms of ameliorating EAE disability [178]. In vitro experiments by this group further showed that glibenclamide blocked TRPM4 currents and glutamate-mediated cytotoxicity, similar to *Trpm4* genetic deletion.

#### 3.7.2. MS and EAE—Clinical Studies

##### MS—Human Expression Patterns

Similar SUR1 ± TRPM4 expression patterns described in the murine EAE models (Section 3.7.1) have been demonstrated in human tissue from MS patients [98,178]. TRPM4 expression in human brain MS samples (13 patients) revealed significantly more TRPM4^+^ axons in active demyelinating lesions vs. controls (*p* = 0.006) [178]. As with the murine findings, TRPM4^+^ axons were irregular and often fragmented; uninflamed peri-plaque white matter (or inactive lesions) did not demonstrate significant increases in TRPM4 axonal expression. Neuronal expression was minimal. Subsequent work independently confirmed these findings and expanded the understanding of the role of SUR1-TRPM4 in MS using autopsy tissue from nine patients (and six controls) [98]. Again, no SUR1 labeling was noted in control white matter. However, in both active and chronic-active white matter lesions, widespread SUR1 immunolabeling was noted in astrocyte-like GFAP^+^ stellate cells, with the highest intensity noted in chronic-active lesions. Co-expression of SUR1-TRPM4 in these stellate cells was predominantly in the perivascular endfeet, and physical co-assembly was demonstrated using immunoFRET analysis. This study also demonstrated that SUR1-TRPM4-expressing astrocytes within human MS lesions also co-expressed other pathogenic molecules, including NOS2, BAFF, and CCL2, similar to the murine findings discussed above (Section 3.7.1).

##### MS—Clinical Trials

There are currently no reported or ongoing clinical trials of SUR1-TRPM4 inhibition in MS.

### 3.8. Neuro-Oncology

Cerebral edema is a frequent offender in the neurological morbidity and mortality caused by both primary brain tumors as well as metastatic CNS disease. Steroids remain a mainstay of treatment, although their use is accompanied by significant side-effects. Understanding the potential contributions of SUR1-TRPM4 to peritumoral edema may be valuable; currently there are two reports exploring this process.

#### 3.8.1. Neuro-Oncology—Preclinical Studies

##### Neuro-Oncology—In Vivo Expression Patterns

In a rat model of metastatic brain tumor, where small cell lung carcinoma or melanoma cells were intracerebrally implanted in nude rats, SUR1 protein expression was shown to be significantly increased compared with the contralateral basal ganglia (*p* < 0.05), measured by immunofluorescence and western blot [41]. Expression was increased in the peritumoral region in both glial and endothelial cells.

##### Neuro-Oncology—In Vivo Channel Blockade

In the same study and models described above (Section 3.10.1), treatment with glibenclamide (4.8 μg loading dose plus 400 ng/h continuous infusion) was compared with dexamethasone (0.35 mg loading dose plus 3 mg/h continuous infusion) [41]. Interestingly, serum glucose measurements did not vary between the two groups, despite the clinical association of steroids being associated with hyperglycemia, and the potential opposite possibility with glibenclamide. Blood–tumor barrier permeability as measured by T1 post-contrast MRI sequences was significantly reduced after treatment with glibenclamide compared with control rats with small cell lung carcinoma metastases (*p* < 0.01), or melanoma metastases (*p* < 0.05). Similar findings were obtained with dexamethasone for small cell lung carcinoma. However, the difference vs. controls was not significant for melanoma. In both metastatic models, rats treated with glibenclamide had less ZO-1 gap distances vs. controls (*p* < 0.01). Despite the benefit on tumors, no differences were identified in median overall survival between vehicle, glibenclamide, or dexamethasone in either tumor model.

#### 3.8.2. Neuro-Oncology—Clinical Studies

##### Neuro-Oncology—Human Expression Patterns

SUR1 expression has been demonstrated in post-mortem brain tissues from both adult and pediatric brain tumors [39]. SUR1 expression was present in all specimens from patients with IDH1 wild type glioblastoma (6 patients), cerebral metastases (12 patients), medulloblastoma (11 patients), supratentorial ependymoma (9 patients), and posterior fossa ependymoma (8 patients). As a percentage of total tissue area, the highest SUR1 expression was found in supratentorial ependymoma and medulloblastomas specimens. Depending on tumor subtype, SUR1 expression was variable in endothelial cells vs. glial cells vs. neurons; in glioblastomas, metastases, and posterior fossa ependymoma specimens, SUR1 expression had the greatest co-localization with NeuN^+^ neurons.

##### Neuro-Oncology—Clinical Trials

There are currently no reported or ongoing clinical trials of SUR1-TRPM4 inhibition in either primary brain tumors or metastatic CNS disease.

### 3.9. Acute Liver Failure (ALF)

Cerebral edema remains a significant cause of morbidity and mortality in ALF, with particular pathology attributed to astrocyte swelling. There is currently one preclinical report of SUR1-TRPM4 and inhibition with glibenclamide in this disease process [186].

#### 3.9.1. ALF—Preclinical Studies

##### ALF—In Vitro Studies

In cultured astrocytes exposed to ammonia, SUR1 protein and *Abcc8* mRNA were both markedly increased vs. controls [186]. Protein expression peaked at 24 h, whereas mRNA expression had a bimodal peak (3 and 24 h). Ammonia-treated astrocytes had increased NF-κB activation (which was reduced by the NF-κB inhibitor BAY11-7082), suggesting that this mechanism may be relevant to astrocytic edema in ALF. In vitro, glibenclamide treatment (5–200 nM) reduced 24 h cell volume, as measured by the 3-O-methyl-[^3^H]-glucose method.

##### ALF—In Vivo Expression Pattern

An in vivo rat model of ALF using thioacetamide treatment (TAA) by the same investigators demonstrated 1.5× SUR1 mRNA upregulation by 48 h. SUR1 immunofluorescence in these mice was the most intense in cortical astrocytes.

##### ALF—In Vivo Channel Blockade

In the same study (Section 3.10.1), treatment of rats with 3 μg/kg (bodyweight) glibenclamide reduced brain water content by 48.7%. Moreover, the increased SUR1 expression noted after TAA-induced ALF was attenuated by glibenclamide. Clinical grading of rat encephalopathy was also improved after glibenclamide treatment. Thus, early data from this single study suggest that this pathway may be relevant to ALF-related cerebral edema, and may warrant further evaluation.

#### 3.9.2. ALF—Clinical Studies

There are currently no clinical studies of SUR1-TRPM4 and/or channel inhibition in ALF.

### 3.10. Status Epilepticus

Although cerebral edema after status epilepticus is less well studied than some of the aforementioned acute neurological conditions, it is an important cause of intracranial hypertension, morbidity, and even mortality in neuro-ICUs [187,188]. There is currently one preclinical study evaluating SUR1-TRPM4 and glibenclamide in an in vivo model of status epilepticus, the results of which suggest that inhibition is beneficial in treating cerebral edema as well as improving outcomes [188]. Additional studies may be informative.

#### 3.10.1. Status Epilepticus–Preclinical Studies

##### Status Epilepticus—In Vivo Expression Patterns

In a rat model of status epilepticus, SUR1 and TRPM4 protein and *Abcc8/Trpm4* mRNA expression was increased within 6 h and persisted for 3 and 7 days, respectively [188]. On day 3, most SUR1 and TRPM4-positive cells localized to the hippocampal CA1 region and piriform cortex; co-labeling indicated expression in both neurons (NeuN) and endothelial cells (vWF).

##### Status Epilepticus—In Vivo Channel Blockade

In the same study (Section 3.10.1), glibenclamide treatment reduced SUR1 and TRPM4 upregulation 3 days after status epilepticus, including expression in the CA1 region and piriform cortex. Neuronal and dendritic loss in these regions was also mitigated with glibenclamide. Expectedly, glibenclamide did not affect seizure susceptibility, but both cerebral edema (brain water content), spontaneously recurring seizures, and mortality were significantly reduced. Moreover, cognitive neurological function was better in glibenclamide-treated rats. Gene knockdown of *Trpm4* had similar effects, with decreased BBB disruption and neuronal loss.

#### 3.10.2. Status Epilepticus—Clinical Studies

There are currently no clinical studies of SUR1-TRPM4 and/or channel inhibition in status epilepticus.

## 4. SUR1-TRPM4 Expression and Inhibition in Other Neurological Conditions

### 4.1. Neuropathic Pain

In neuropathic pain, responses to innocuous as well as noxious stimuli are magnified, and pain may also occur spontaneously [189]. After peripheral nerve injury (PNI), both peripheral and central inflammation may modulate neuropathic pain and influence pain hypersensitivity [190,191,192]. Given the overlay between pain and inflammation, the role of SUR1 ± TRPM4 has been explored in preclinical models of chronic and neuropathic pain (Table 7) [69,121]. If this targeted therapy is able to clinically translate, it may provide an alternative, non-addictive approach towards neuropathic pain.

#### 4.1.1. Neuropathic Pain—Preclinical Studies

##### Neuropathic Pain—In Vivo Expression Patterns

In a model of neuropathic pain, spinal cord tissues were harvested from wildtype mice 3–14 days after unilateral sciatic nerve cuffing. Quantitative analysis of SUR1 immunopositivity demonstrated a progressive increase in protein expression in the ipsilateral dorsal horn over 14 days, with minimal SUR1 labeling in normal wildtype controls [69]. The majority of SUR1 expression was in astrocytes (GFAP^+^ stellate cells), with minimal co-localization of SUR1 with markers of other cell types such as oligodendrocytes or microglia. Double immunolabeling showed co-localization of SUR1-TRPM4 within astrocytes, and proximal ligation assays confirmed co-assembly of these heteromers in the dorsal horn [69].

##### Neuropathic Pain—In Vivo Channel Blockade

Several studies have evaluated the potential analgesic effects of glibenclamide and other SUR agonists/antagonists in conjunction with other analgesics vis-à-vis K_ATP_ channels in neuropathic pain [193,194,195,196,197,198,199,200]. In these studies, single doses of glibenclamide did not influence pain thresholds but they did blunt the anesthetic effects of the analgesic being tested. Thus, K_ATP_ channels may be important to augment the analgesic effect of other compounds, but seem to be less important in basal conditions of neuropathic pain. In a chronic pain murine model of spinal nerve ligation (SNL) without other concurrent analgesics, *Abcc8* (SUR1) and *Kcnj11* (KIR6.2) mRNA was downregulated in the dorsal root ganglia and sciatic nerves 2 months after injury [121]. Administration of the SUR1 agonist diazoxide by intraplantar or intrathecal injection alleviated mechanical hypersensitivity after SNL. Mice lacking the SUR1 K_ATP_ channel subunit due to genetic modification, either by global knockout (*Abcc8* KO) or intrathecal injection of short hairpin RNA (*Abcc8* shRNA), displayed mechanical hypersensitivity when compared to wild type and control mice [121]. The role of SUR1-TRPM4, however, was not explored in this study.

Recently, neuropathic pain was induced by unilateral sciatic nerve cuffing in wild type mice and mice with global or *p-Gfap-cre- or pGFAP-cre*/ERT2-driven *Abcc8*/SUR1 deletion or global *Trpm4* deletion [69]. Wild type mice received prophylactic (starting on post-operative day [pod]-0) or therapeutic (starting on pod-21) administration of the SUR1 antagonist, glibenclamide (10 mg IP) daily. Wild type mice with sciatic nerve cuffing showed pain behavior (mechanical allodynia, thermal hyperalgesia) and newly upregulated SUR1-TRPM4 expression in dorsal horn astrocytes (Section 3.8.1). Global and *p-Gfap-cre*-driven *Abcc8* deletion as well as global *Trpm4* deletion prevented pain behaviors from developing. *Abcc8* silencing (by tamoxifen) in *p-GFAP-cre*/ERT2 mice, after pain behaviors had developed, was found to be beneficial, with continued improvement noted over the subsequent 14 days. Glibenclamide (both prophylactically and after allodynia was established) was similarly beneficial in terms of preventing or reducing allodynia. Glibenclamide administration was also associated with reduced astrocytic expression of IL-6, CCL2 and CXCL1, consistent with reduced neuroinflammation in the dorsal horn [69].

#### 4.1.2. Neuropathic Pain—Clinical Studies

##### Neuropathic Pain—Human Expression Patterns

There are currently no reported studies of human SUR1-TRPM4 expression in neuropathic pain.

##### Neuropathic Pain—Clinical Trials

There are currently no reported or ongoing clinical trials of SUR1-TRPM4 inhibition in neuropathic pain.

### 4.2. HIV-Associated Neurocognitive Disorders (HAND)

HIV-associated neurocognitive disorders (HAND) are characterized by glial activation, cytokine/chemokine dysregulation, and neuronal damage, thought to be a result of chronic neuroinflammation and neurotoxicity [201]. It has been speculated that HIV-1 viral protein R (Vpr), may contribute to HAND. A recently published single study has evaluated the role of SUR1-TRPM4 expression patterns and inhibition in a preclinical mouse models of HIV (Tg26) as well as postmortem human tissue [201].

#### 4.2.1. HAND—Preclinical Studies

##### HAND—In Vivo Expression Patterns

In Tg26 mouse brain, HIV-1 infection increased several inflammatory markers in the hippocampus vs. wildtype controls [201]; these markers are known to upregulate SUR1 including TLR4, TNFa, and NF-κB (Section 2.4). This was also associated with upregulated expression of astrocytic SUR1-TRPM4 in the hippocampus of infected mice (*p* < 0.001 vs. wildtype controls). Elevated SUR1 protein expression correlated with higher *Abcc8* mRNA levels.

##### HAND—In Vitro Channel Blockade

The benefits of glibenclamide in terms of blocking Vpr-induced apoptosis have been demonstrated in vitro, using cells from the human glioblastoma SNB19 cell line (of glial cell origin) [201]. In this study, a concentration-dependent inhibition of Vpr-induced apoptosis was observed at 2, 4 and 6 h post-infection using glibenclamide at increasing concentrations of 10, 30 and 90 mM.

#### 4.2.2. HAND—Clinical Studies

##### HAND—Human Expression Patterns

Similar to the murine findings (Section 3.9.1), in post-mortem brain tissue from HIV-infected patients, HIV-1 and Vpr expression was accompanied by increased expression of both SUR1 and TRPM4 in hippocampal astrocytes (*p* < 0.01 vs. non-infected patients) [201]. The two channel subunits co-localized, consistent with channel upregulation. Overexpression of other inflammatory markers such as TLR4, NF-κB, and TNFa was also noted in hippocampal astrocytes, although the difference between control vs. HIV infected NF-κB did not reach statistical significance. In vitro, in human glioblastoma SNB19 cells, expression of HIV-1 Vpr was also associated with increased SUR1 expression, as well as increased expression of the same set of proinflammatory markers [201].

##### HAND—Clinical Trials

There are currently no reported or ongoing clinical trials of SUR1-TRPM4 inhibition in HAND.

### 4.3. Retinopathy

Retinopathy can occur as a consequence of several different processes, with underlying mechanisms including neuroinflammation, metabolic disturbances, and possibly vasogenic edema. Indeed, diabetic retinopathy is a major cause of vision loss worldwide, and may be related to SUR1 regulation of both KIR6.2 as well as TRPM4. Although the data are limited to a single study, they support a potential role for SUR1 in excitotoxic retinal cell death, as well as a benefit of glibenclamide—as demonstrated in several rodent models [202].

#### 4.3.1. Retinopathy—Preclinical Studies

##### Retinal Preclinical SUR1 Expression Patterns

SUR1 expression was demonstrated in non-human primate retinas with similar patterns noted in non-diabetic humans (Section 4.3.2) [202]. Expression was noted in all three retinal layers: ganglion cell layer (GCL), outer nuclear layer (ONL), and outer limiting membrane (OLM). SUR1 was enriched in the macula, and was also expressed in glial Müller cells and retinal vessels. In the fovea (responsible for visual acuity), SUR1 was expressed in the cone photoreceptors (blue, red/green) and the outer limiting membrane. In most cell types, particularly Müller glial cells and cones, SUR1 colocalized with both TRPM4 and KIR6.2.

##### Retinopathy—In Vivo Channel Blockade

In three rat models of retinopathy (adult diabetic retinopathy, NMDA-induced retinal excitotoxicity, and neonatal hyperglycemia-induced retinal neurodegeneration), intraocular glibenclamide improved retinal structure and function. Glibenclamide reduced cellular edema (Müller cells) and restored photoreceptor morphology without affecting glucose levels. Glibenclamide treatment also reduced apoptosis. These neuroprotective effects were abolished with pretreatment with siRNA directed against SUR1. Transcriptomic analysis revealed mRNA upregulation of known neuroprotective and antioxidant genes including lipocalin-2, osteopontin, clusterin, C1q and Bcl2a in glibenclamide-injected rat retinas.

#### 4.3.2. Retinal Expression—Clinical Studies

In the same study as above (Section 4.3.1), human ocular tissue was obtained from a non-diabetic patient after enucleation for an untreated anterior uveal melanoma. The posterior retina was intact. In this patient, SUR1 was expressed in all retinal layers (GCL, OLM, ONL), retinal Müller glial cells, cone photoreceptors, and retinal vessels. It colocalized partially with TRPM4 in human retinal astrocytes in the GCL, OLM, and cone photoreceptors. KIR6.2 was also expressed in the inner nuclear layer, OLM, Müller cells, and around retinal vessels—but not in retinal astrocytes. Systematic study of SUR1 ± TRPM4 ± KIR6.2 expression in the healthy vs. diseased human retina has not yet been reported.

## 5. Conclusions

Advances over the past decade have broadened and deepened our understanding of the emerging role of SUR1-TRPM4 as a unifying mediator of secondary injury across a spectrum of ostensibly different CNS pathologies. De novo upregulation of this channel in all cells of the neurovascular unit has been shown to contribute to cerebral edema, hemorrhage progression, neuroinflammation and/or cell death in preclinical models of ischemic stroke, TBI, SCI, cardiac arrest, SAH, ICH, MS/EAE, neuropathic pain, HAND, and brain tumors. Patterns of channel expression in human patients with these diseases mimic those reported in rodent models. Across these conditions, genetic silencing or pharmacological inhibition of this channel with drugs—notably glibenclamide—has conferred protection, as shown in a growing body of preclinical models and early clinical trials. Findings have been robust and generally consistent across several independent laboratories despite the study nuances reviewed here. Although nature is notoriously conservative, it is rare for a uniquely upregulated molecular target to show such exciting potential, spanning not only acute ischemic and hemorrhagic CNS diseases, but also acute and chronic inflammatory CNS diseases.

The preponderance of research on SUR1-TRPM4 in CNS injury has been in ischemic stroke and TBI. Those findings have fueled the recent and ongoing clinical trials of SUR1-TRPM4 inhibition with BIIB093, an intravenous formulation of glibenclamide. The phase II clinical trial of BIIB093 in large hemispheric infarction (GAMES-RP) yielded promising results. Both CHARM (phase III trial in large hemispheric infarction) and ASTRAL (phase II trial in contusional TBI) are enrolling, and their results are eagerly awaited. Future research on pharmacodynamic response, theragnostic biomarkers (including imaging, serum, CSF), endophenotyping intermediate/mechanistically relevant outcomes, and genetic variation in this pathway may further optimize the ability to harness this treatment by optimizing timing, dosing, or risk-stratification. This may be particularly important given the historical context of the struggle to translate neuroprotectants. While multifactorial, this has largely been attributed to challenges of patient and disease heterogeneity compounded by treatment homogeneity and lack of informative biomarkers for predictive and prognostic enrichment. The suggestion of genetic underpinnings driving the impact of the SUR1-TRPM4 pathway in different patients suggests that this may also represent a special opportunity to direct future precision-trial design.

Despite the ubiquitous devastation of secondary injury processes such as severe cerebral edema and hemorrhage progression across different types of acute brain injury, treatment options remain limited. Although therapies such as hyperosmolar agents and craniectomy may be lifesaving, they are reactive treatments only equipped only to deal with the downstream consequences of cerebral edema after it has already occurred, rather than addressing the root-cause. SUR1-TRPM4 is newly upregulated after CNS injury and may be pivotal in generating cerebral edema and hemorrhage progression. It also interfaces with several other key mediators of these secondary injury processes as well as neuroinflammation and cell death. This provides a unique opportunity to limit the development of cerebral edema. Based on the pathophysiology and mounting evidence in both preclinical models and clinical studies, targeting this channel has the potential for successful translation that would improve patients’ lives after CNS injury.

## Figures and Tables

**Figure 1 ijms-22-11899-f001:**
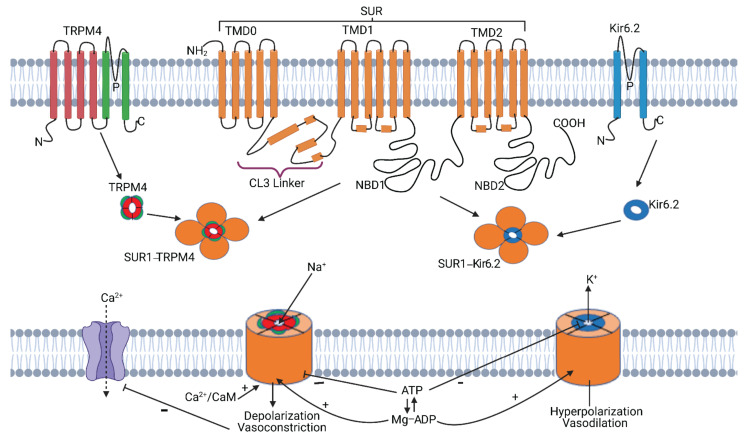
Schematic depiction of SUR1, TRPM4, and Kir6.2 topology. TRPM4 monomers consist of six transmembrane helices with a pore region (P). SUR1 is composed of two interacting six-helix trans-membrane domains (TMD1 and TMD2), each containing a nucleotide binding domain NBD1 and NBD2. TMD0 interacts directly with Kir6.2. Four TRPM4 subunits oligomerize with four SUR1 subunits to form the functional SUR1-TRPM4 octameric channel. Influx of Na^+^ via the nonselective monovalent cation pore-forming subunit TRPM4 leads to depolarization. Figure created with BioRender.com (2021).

**Figure 2 ijms-22-11899-f002:**
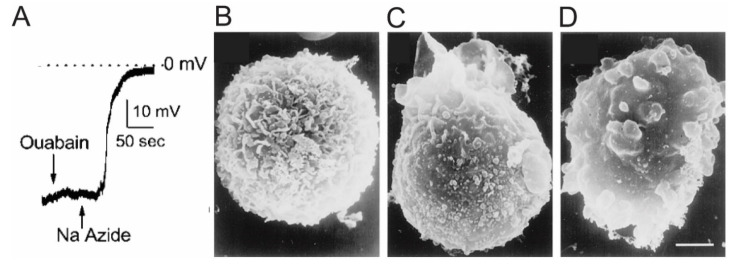
(**A**) Sodium azide induced ATP depletion, resulting in cellular depolarization. Current clamp recording demonstrating cellular resting potential (~60 mV). After exposure to 1 mM oubain (downward arrow), cell depolarization was <5 mV with rapid recovery after washout. Conversely, 3 min exposure to 1 mM sodium azide induced ATP depletion and resulted in a depolarizing inward current with cell depolarization to almost 0 mV. Scanning electron micrograph of a freshly isolated reactive astrocyte under control conditions (**B**), 5 min after exposure to 1 mM sodium azide (**C**) and 25 min after exposure to 1 mM sodium azide (**D**). Scale bar is 12 μm. Adapted with permission from Ref. [7].

**Figure 3 ijms-22-11899-f003:**
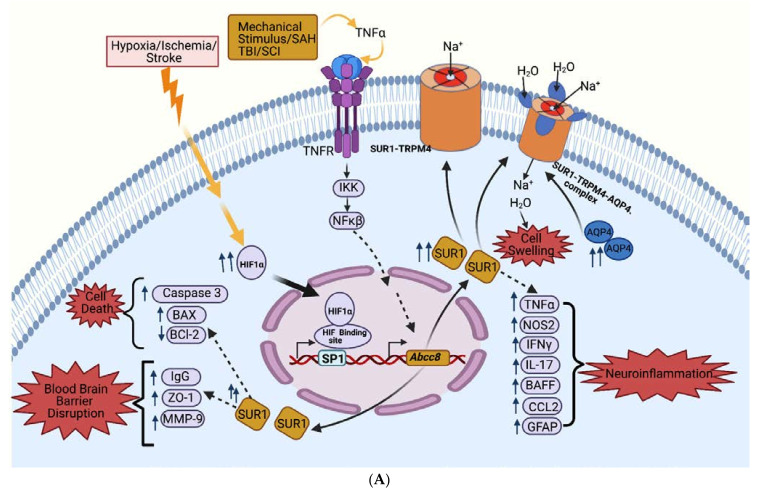
(**A**) Schematic depiction of upregulated pathways after CNS injury related to SUR1-TRPM4. Hypoxia/ischemia results in upregulation of HIF1α, which is transported to the nucleus where it binds with the HIF-binding site, leading to the upregulation of SP1—which in turn leads to the upregulated transcription of *Abcc8* (SUR1). Mechanical stimuli after traumatic brain injury or spinal cord injury results in increased levels of TNFα, which is an upstream regulator of *Abcc8* (SUR1). Increased levels of TNFα lead to upregulation of IKK and NF-κB, ultimately resulting in transcriptional activation of *Abcc8* (SUR1). Upregulation of SUR1 leads to increased association of SUR1-TRPM4 on the plasma membrane. An open SUR1-TRPM4 channel results in Na^+^ influx, leading to depolarization and an oncotic gradient. Association of the SUR1-TRPM4-AQP4 complex results in increased influx of water into the cell (following the oncotic gradient), ultimately leading to cell swelling. Increased SUR1 may contribute to neuroinflammation (by activating NOS2, INFγ, IL-17, BAFF, CCL2, GFAP), disruption of the blood brain barrier (by activating IgG, MMP-9 and ZO-1), and cell death (via upregulation of caspase 3 and BAX). (**B**) Schematic representation of the signaling effects of SUR1 inhibition via genetic *Abcc8* knockout or pharmacological inhibition by glibenclamide. Edema is reduced via channel blockade. Neuroinflammation is decreased (lower levels of TNFα, NOS2, INFγ, IL-17, BAFF, CCL2, GFAP); the blood–brain barrier disruption is minimized (downregulated ZO-1, MMP-9), and cell death reduced (upregulation BCL-2, downregulation of BAX and caspase 3). Figure created with BioRender.com (2021).

**Figure 4 ijms-22-11899-f004:**
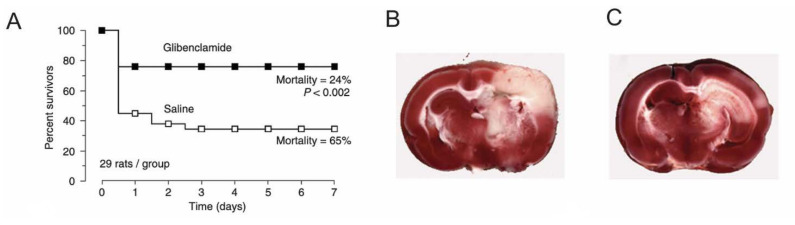
(**A**) Graph showing percent survival of rats over 7 days after middle cerebral artery occlusion (MCAO, malignant cerebral edema model) when treated by saline (open squares) vs. glibenclamide (filled black squares). At 7 days, mortality was significantly different in the glibenclamide group (24%) vs. vehicle group (65%, *p* < 0.002). Representative TTC-stained coronal sections 2 days after MCAO (thromboembolic model) in a rat treated with saline (**B**) vs. glibenclamide (**C**) showing cortical sparing in the latter. Adapted with permission from Ref. [65].

**Figure 5 ijms-22-11899-f005:**
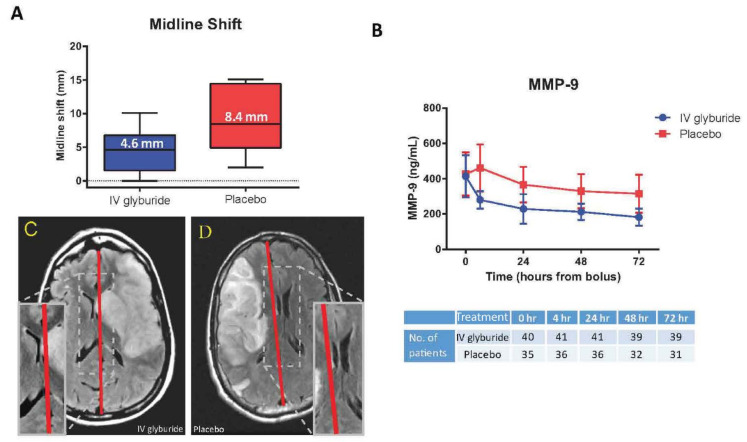
Figures from the GAMES-RP clinical trial demonstrating the effect of glyburide on midline shift (**A**,**C**,**D**) and plasma MMP-9 levels (**B**). (**A**) Boxplot showing the median midline shift (horizontal line, box = interquartile range, whiskers = 10–90 percentile) in each treatment group in the per-protocol sample. (**B**) Temporal profile of mean total plasma MMP9 levels for the per-protocol sample (error bars are 95% confidence intervals at the specific timepoint shown). (**C**,**D**) Representative examples of median extent of midline shift on follow-up brain MRI with a patient treated with intravenous glyburide (5 mm) vs a placebo-treated patient (9 mm). The redline is a reference showing the brain midline. Adapted with permission from Ref. [47].

**Figure 6 ijms-22-11899-f006:**
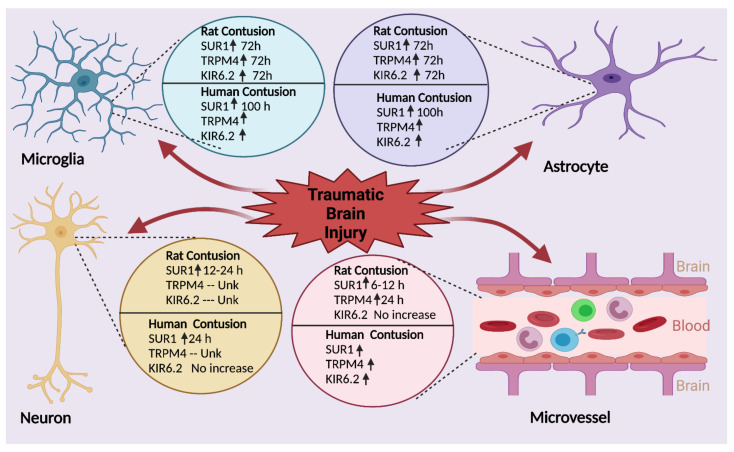
Schematic representation of SUR1, TRPM4, and KIR6.2 expression and timelines in neuron, microglia, astrocytes, and microvessels after TBI in rodents and humans. Figure created with BioRender.com (2021).

**Figure 7 ijms-22-11899-f007:**
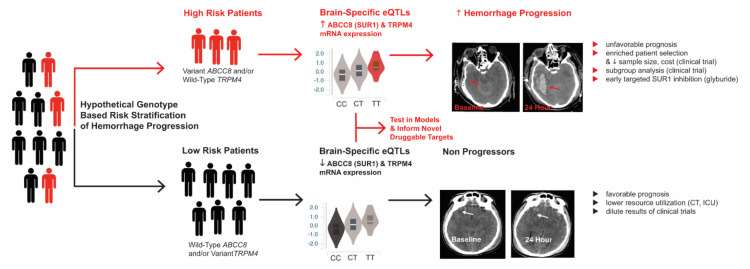
Hypothetical schematic example of precision medicine using genotype-based risk stratification of hemorrhage progression. Patients with high-risk genotypes of *ABCC8* or *TRPM4* variants (red) may have higher levels of channel mRNA and/or protein expression and in turn be at higher risk of hemorrhage progression after traumatic brain injury. Genotype-based stratification could inform clinician prognostication, enrich patient selection/decrease sample size and cost of clinical trials, guide subgroup analyses, and also be valuable for discovery of novel therapeutic targets based on gene expression/regulation/post translational modifications in this pathway.

**Table 1 ijms-22-11899-t001:** Key SUR1 signaling pathways.

Study Title	Disease and Model Details	Regulation/Inhibition/Interaction	Signaling Pathway Proteins	Authors/Year
Glibenclamide reduces inflammation, vasogenic edema, and caspase-3 activation after subarachnoid hemorrhage	SAH (rat, filament puncture)	Inhibition of SUR1 by GLI	Treatment ↓ IgG extravasation Treatment ↓ ZO-1 expression and prevention of redistribution with significant ↓ in cytoplasmic localization and restored intracellular localization Treatment ↓ endothelial caspase 3 expression Treatment ↓ GFAP expression	Simard et al., 2009 [40]
Inhibition of the Sur1-Trpm4 channel reduces neuroinflammation and cognitive impairment in subarachnoid hemorrhage	SAH (rat, filament puncture and entorhinal injection)	Gene suppression of *Abcc8*/Inhibition of SUR1 by GLI	Inhibition/treatment ↓ TNFα expression	Tosun et al., 2013 [36]
Glibenclamide Attenuates Blood–Brain Barrier Disruption in Adult Mice after Traumatic Brain Injury	TBI (mouse, CCI)	Inhibition of SUR1 by GLI	Treatment ↓ AQP4 expression Treatment ↓ ZO-1 expression Treatment ↓ BAX expression Treatment ↑ Bcl-2 expression	Xu et al., 2017 [44]
SUR1-TRPM4 channel activation and phasic secretion of MMP-9 induced by tPA in brain endothelial cells	Cerebral ischemia/ reperfusion (in vitro, and rat ischemia-reperfusion)	NF-κB activation/ SUR1-TRPM4 channel opening by tPA	De novo expression of SUR1 ↓ phasic MMP9 secretion	Gerzanich et al., 2018 [96]
Glibenclamide reduces hippocampal injury and preserves rapid spatial learning in a model of traumatic brain injury	TBI (rat, cortical impact)	Inhibition of SUR1 by GLI	Treatment ↓ caspase 3 expression	Patel et al., 2010 [30]
Silencing of *Abcc8* or inhibition of newly upregulated Sur1-Trpm4 reduce inflammation and disease progression in experimental autoimmune encephalomyelitis	EAE (mouse EAE, MOG_33–55_ peptide)	*Abcc8*^−/−^ mice/Inhibition of SUR1 by GLI	Treatment ↓ TNFα, IFNγ, IL-17 expression	Makar et al., 2015 [97]
Salutary effects of glibenclamide during the chronic phase of murine experimental autoimmune encephalomyelitis	EAE (mouse EAE, MOG_33–55_ peptide)	Inhibition of SUR1 by GLI	Treatment ↓ TNFα, BAFF, CCL2, NOS2 expression	Gerzanich et al., 2017 [98]
Glibenclamide pretreatment protects against chronic memory dysfunction and glial activation in rat cranial blast traumatic brain injury	TBI (rat, direct cranial blast)	Inhibition of SUR1 by GLI	Treatment ↓ GFAP and Iba-1 expression	Stokum et al., 2017 [45]
The Sur1-Trpm4 channel regulates NOS2 transcription in TLR4-activated microglia	Activation of TLR4 by LPS (primary culture, adult rat microglia)	*Abcc8*^−/−^ mice/Inhibition of SUR1 by GLI	Treatment ↓ NOS2 expression	Kurland et al., 2016 [92]

**Table 2 ijms-22-11899-t002:** Pre-clinical and clinical studies evaluating SUR1 inhibition for ischemic stroke.

Authors, Year	Study Title	Model	Species	Drug Details	Results
In vitro studies: ischemic stroke					
Chen et al., 2001 [7]	Cell swelling and a nonselective cation channel regulated by internal Ca^2+^ and ATP in native reactive astrocytes from adult rat brain	Chemically induced hypoxia in non-reactive isolated astrocytes using 1 nM NaN_3_	NA	NA	↓ cytosolic ATP leads to activation of SUR1-TRPM4 channels and cell membrane depolarization SUR1-TRPM4 activation leads to influx of Na^+^, water, resulting in cell swelling and blebbing
Chen et al., 2003 [8]	Functional coupling between sulfonylurea receptor type 1 and a nonselective cation channel in reactive astrocytes from adult rat brain	Chemically induced hypoxia in non-reactive astrocytes using 1 nM NaN_3_	NA	GLI	SUR1 is functionally coupled to the pore-forming portion of the NC_Ca-ATP_ channel Inhibition of SUR1 by GLI prevented cell blebbing
In vivo studies: ischemic stroke					
Simard et al., 2006 [65]	Newly expressed SUR1-regulated NC(Ca-ATP) channel mediates cerebral edema after ischemic stroke	Massive middle cerebral artery (MCA) infarction associated with malignant cerebral edema	Rat	GLI Infusion rate: 75 ng/h Infusion time: 2–3 min after MCAo	↑ SUR1 level in the infarct core within 2–3 h of MCAO in neuron and capillary which declined by 8 h In peri-infarct area SUR1 levels persisted for 8–16 h GLI prevented Na^+^ influx, cell depolarization and cell blebbing resulting in ↓ cerebral edema GLI i infarct volume at 2 d and 7 d
Simard et al., 2009 [112]	Protective effect of delayed treatment with low-dose glibenclamide in three models of ischemic stroke	3 models Thromboembolic MCA occlusion (MCAo), Transient MCAo and Permanent MCAo	Rat	GLI Time: variable Loading dose: 3.3–10 mg/kg Infusion rate: 75–200 ng/h	In thromboembolic MCAo at 6 h, GLI treatment i total lesion volume by 53% GLI ↓ cortical lesion volume by 41% in transient MCAo at 5.75 h GLI ↓ cortical lesion volume by 51% in permanent MCAo at 4 h
Simard et al., 2010 [113]	Glibenclamide is superior to decompressive craniectomy in a rat model of malignant stroke	MCAo severe ischemia reperfusion	Rat	GLI Time: 15 min before MCAo Loading dose: 10 mg/kg i.p. Infusion rate: 200 ng/h	6 h after onset of ischemia, GLI treatment was as effective as immediate decompressive craniectomy in ↓ mortality
Abdallah et al., 2011 [114]	Glibenclamide ameliorates ischemia-reperfusion injury via modulating oxidative stress and inflammatory mediators in the rat hippocampus	Ischemia reperfusion	Rat	GLI Time: 10 min before ischemia/reperfusion Dose: 1 mg/kg i.p.	GLI treatment ↓ neutrophil recruitment, ameliorated TNFα, prostaglandin E2 and boosted anti-inflammatory cytokines
Simard et al., 2012 [115]	Glibenclamide 10 h treatment window in a clinically relevant model of stroke.	Intra-arterial occluder MCAo	Rat	GLI Time: 4.5 vs. 10 h post MCAo Loading dose: 10 μg/kg, Infusion rate: 200 ng/h	GLI treatment improved hemispheric swelling, neurological outcome and 48 h mortality
Ortega et al., 2012 [116]	ATP-dependent potassium channel blockade strengthens microglial neuroprotection after hypoxia-ischemia in rats	Transient MCAo	Rat	GLI Time: 6, 12, 24 h after MCAo Doses: 0.06 μg, 0.6 μg, 6 μg	GLI treatment improved neurological score and preserved neurons in the lesioned core 3 d after reperfusion
Wali et al., 2012 [117]	Glibenclamide administration attenuates infarct volume, hemispheric swelling, and functional impairments following permanent focal cerebral ischemia in rats	Permanent MCAo	Rat	GLI Time: 5 min after MCAo Loading dose: 10 mg/kg, Infusion rate: 200 ng/h	GLI treatment significantly ↓ infarct volume, hemispheric swelling, grip strength and lowered neurological severity scores
Ortega et al., 2013 [118]	Glibenclamide enhances neurogenesis and improves long-term functional recovery after transient focal cerebral ischemia	Transient MCAo	Rat	GLI Time: 6 h, 12 h, 24 h after reperfusion Dose: 0.6 mg	Migration of double cortin-positive cells were observed within 72 h of GLI treatment GLI treatment continued neurogenesis even after 30 d of MCAo, and also yielded both cognitive and sensorimotor improvement
Arikan et al., 2017 [119]	Malignant infarction of the middle cerebral artery in a porcine model. A pilot study	Malignant infarction of the middle cerebral artery	Pig	NA	↑ SUR1-TRPM4 in core, penumbra and contralateral neurons and microvessels 5 h after ischemia
Alquisiras et al., 2020 [120]	Resveratrol reduces cerebral edema through inhibition of de novo SUR1 expression induced after focal ischemia	MCAo	Rat	NA	Resveratrol treatment ↓ *Abcc8* mRNA and SUR1 protein expression Resveratrol treatment ↓ cerebral edema, BBB disruption, AQP4 expression and improved neurological outcome and survival (by ~40%)
Woo et al., 2020 [121]	SUR1-TRPM4 channels, not K_ATP_, mediate brain swelling following cerebral ischemia	Permanent MCAo	Rat	*Abcc8* antisense ODN *Trpm4* antisense ODN	↓ hemispheric swelling
Human expression studies: ischemic stroke					
Mehta et al., 2013 [122]	Sulfonylurea receptor 1 expression in human cerebral infarcts	Post-mortem brain specimen within 31 d after ischemic stroke	13 patients	NA	SUR1 expression was ↑ in all cases in both neurons and endothelial cells and was detected as early as 24 h and persisted for 7–10 d
Mehta et al., 2015 [68]	Sur1-Trpm4 cation channel expression in human cerebral infarcts	Post-mortem brain specimen	14 patients	NA	TRPM4 ↑ and persisted even at 1 m post-infarct FRET analysis showed co-localization of TRPM4 with SUR1 in neurons, endothelial cells and astrocytes
Clinical retrospective studies: ischemic stroke					
Kunte et al., 2007 [123]	Sulfonylureas improve outcome in patients with type 2 diabetes and acute ischemic stroke	Acute ischemic stroke	61 patients	Oral sulfonylurea	Sulfonylurea treatment leads to >4 point reduction in NIHSS score in 36.4% patients (vs. 7.1% in untreated) Sulfonylurea treatment showed an improved modified Rankin scale score of ≤2
Favilla et al., 2011 [124]	Sulfonylurea use before stroke does not influence outcome	Patients enrolled in non-reperfusion ischemic stroke trials	1050 patients	Oral sulfonylurea	Post-stroke sulfonylurea treatment showed a trend toward benefit
Kunte et al., 2012 [125]	Hemorrhagic transformation of ischemic stroke in diabetics on sulfonylureas	Diabetic patients with acute ischemic stroke	220 patients	Oral sulfonylurea	Sulfonylurea i post-stroke symptomatic hemorrhagic transformation and mortality
Horsdal et al., 2012 [126]	Type of preadmission antidiabetic treatment and outcome among patients with ischemic stroke: a nationwide follow-up study	Diabetic patient with acute ischemic stroke	4817 patients	Oral sulfonylurea	No association between preadmission use of sulfonylurea use and clinical outcome after ischemic stroke
Clinical trials: ischemic stroke					
Sheth et al., 2014 [117]	Pilot study of intravenous glyburide in patients with a large ischemic stroke	Large hemispheric infarction	10 patients	Intravenous glyburide (RP-1127)	Drug well tolerated without any safety concerns or hypoglycemia
Sheth et al., 2016 [47]	Safety and efficacy of intravenous glyburide on brain swelling after large hemispheric infarction (GAMES-RP): a randomized, double-blind, placebo-controlled phase 2 trial	Large hemispheric infarction	86 patients	Intravenous glyburide (RP-1127) Phase II double-blind (GAMES-RP)	Drug is safe and well tolerated Did not meet primary composite endpoint of mRS 0–4 at 90 d without decompressive craniectomy Improved midline shift Reduced plasma MMP-9 Trend towards survival benefit

**Table 3 ijms-22-11899-t003:** Ongoing Clinical Trials for Glyburide.

NCT Identifier	Phase	Drug	Studied Population	Outcome/References
NCT03741530	Phase I (GATE-ICH)	Oral glyburide (1.25 mg)	ICH	Completed, results not yet reported [127]
NCT02864953	Phase III (CHARM)	BIIB093 (newest name for RP-1127)	Large hemispheric infarction including thrombectomy	Recruiting [46]
NCT03954041	Phase II (ASTRAL)	Intravenous glyburide (RP-1127)	Brain contusion	Recruiting [46]
NCT02524379	Phase I (SCING)	Oral glyburide (3.125–2.5 mg on days 1–3)	Acute cervical traumatic SCI	Active, not recruiting [46]
NCT03569540	Phase IV (GASH)	Oral glyburide (0.5 mg)	SAH	Unknown [128]

**Table 4 ijms-22-11899-t004:** Pre-clinical and clinical studies evaluating SUR1 inhibition for traumatic brain injury (TBI).

Authors, Year	Study Title	Model	Species	Drug Details	Results
In vivo studies: TBI					
Simard et al., 2009 [26]	Key role of sulfonylurea receptor 1 in progressive secondary hemorrhage after brain contusion	Weight-drop model of focal cortical contusion (10 g dropped from 5 cm, velocity = 1 m/s)	Rat	GLI Time: 10 min of injury Loading dose: 10 mg/kg i.p, Infusion rate: 200 ng/h	SUR1 h as early as 3 h after injury. SUR1 upregulation was predominantly h in microvessels and remained elevated up to 24 h After 24 h, SUR1 expression was h in thalamus and hippocampus BBB integrity was maintained with reduced blood extravasation at 3 h, 6 h and maximal benefit at 12 h. Hemorrhage progression was reduced by 45 min in GLI-treated rats
Patel et al., 2010 [30]	Glibenclamide reduces hippocampal injury and preserves rapid spatial learning in a model of traumatic brain injury.	Milder injury weight drop model (10 g dropped from 3 cm, velocity = 0.77 m/s)	Rat	GLI Time: 10 min of injury Loading dose: 10 mg/kg i.p, Infusion rate: 200 ng/h	h SUR1 protein and *Abcc8* mRNA expression in hippocampal neurons at 6 h and peaks at 24 h h Sp1 expression preceded SUR1 expression
Zweckberger et al., 2014 [43]	Glibenclamide reduces secondary brain damage after experimental traumatic brain injury	CCI (1.5mm tissue displacement, velocity = 7.5 m/s, dwell time = 300 ms)	Rat	GLI Time: 15 min after CCI Loading dose 10 mg/kg i.p, Infusion rate 10 mL/h	GLI i brain water content i contusion volume 8 h, 24 h, 72 h and 7 d post-CCI
Xu et al., 2017 [44]	Glibenclamide attenuates blood–brain barrier disruption in adult mice after traumatic brain injury	CCI (1.5 mm tissue displacement, velocity = 1.5 m/s, dwell time = 100 ms)	Mouse	GLI Time: immediately after CCI Dose:10 mg i.p.	GLI i loss of ZO-1 and occludin GLI i brain water content, i BBB permeability, apoptosis (JNK/c-jun signaling)
Jha et al., 2018 [24]	Glibenclamide produces region-dependent effects on cerebral edema in a combined injury model of traumatic brain injury and hemorrhagic shock in mice	CCI (5 m/s, 1 mm depth) + hemorrhagic shock (HS)	Mouse	GLI Time: 10 min after CCI Loading dose: 20 mg/kg i.v. Infusion rate: 0.4 mg/h	Low dose GLI did not reduce ipsilateral edema Low dose GLI normalized contralateral edema in CCI + HS.
Gerzanich et al., 2019 [29]	Sulfonylurea receptor 1, transient receptor potential cation channel subfamily M member 4, and kir6.2: role in hemorrhagic progression of contusion	CCI (4.5 mm tissue displacement, velocity = 1 m/s, dwell time = 200 ms)	Rat	GLI Time: 10 min after CCI Loading dose: 10 mg/kg i.p., Infusion rate: 400 ng/h	SUR1 expression in the core tissue peaked at 6 h, remained elevated at 12 h, decreased at 24 h and upregulated again at 72 h SUR1 expression was h at 24 h predominantly in microvessels but at 72 h SUR1 h in microglia Hemorrhage progression was i by 60% in GLI treated rats
Jha et al., 2021 [33]	Glibenclamide treatment in traumatic brain injury: operation brain trauma therapy	Fluid percussion injury (FPI), CCI (4 m/s; 2.6 mm depth), penetrating ballistic like brain injury	Rat	GLI Time: 10 min after injury Loading dose: 10 mg/kg i.p., Infusion rate: 0.20 μg/h	GLI treatment improved lesion volume and motor function in CCI and did not show any benefit in FPI or penetrating injury
Tata et al., 2021 [35]	*Abcc8* (Sulfonylurea receptor-1) impact on brain atrophy after TBI varies by sex	CCI (1.2 mm displacement, velocity = 5 m/s, dwell time = 50–60 ms)	Mouse	NA (*Abcc8*^−/−^)	*Abcc8* KO mice had smaller contusion volume and larger normalized contralateral (right) hemisphere volumes after injury vs. wild type.
Human expression and genetics studies: TBI					
Martinez et al., 2015 [38]	Sulfonylurea receptor 1 in humans with post-traumatic brain contusions	Contusional TBI	26 patient samples	NA	SUR1 significantly upregulated in all cell types Neuronal SUR1 was detected by 6 h post TBI, with a peak at 24 h In endothelial cells, h SUR1 level persisted up to 100 h
Jha et al., 2017 [24]	Sulfonylurea receptor-1: a novel biomarker for cerebral edema in severe traumatic brain injury	CSF samples from severe TBI	28 patients 15 controls	NA	Level of SUR1 was h in the CSF of all patients ICP trajectory mirrored SUR1 expression after temporal delay SUR1 level, radiographic CT edema and initial degree of intracranial hypertension were associated
Jha et al., 2017 [123]	ABCC8 single nucleotide polymorphisms are associated with cerebral edema in severe TBI	Candidate gene study in severe TBI (*ABCC8*)	385 patients analyzed	NA	Four SNPs rs2283261, rs3819521, rs2283261, rs3819521 were associated with measures of cerebral edema
Jha et al., 2018 [120]	Regionally clustered ABCC8 polymorphisms in a prospective cohort predict cerebral oedema and outcome in severe traumatic brain injury	Tag-SNP study in severe TBI (*ABCC8*)	410 patients analyzed	NA	rs7105832 and rs2237982 were associated with intracranial pressure and radiographic edema rs2237982 was also associated with 3-month GOS Different SNPs (rs11024286, rs4148622) were associated with 3-month GOS. Significant SNPs clustered upstream (opposite end of the gene vs. those significant in disorders of glucose metabolism) Significant SNPs spatially clustered flanking exons encoding the sulfonylurea receptor site and transmembrane domain 0/loop 0 (juxtaposing the channel pore/binding site
Castro et al., 2019 [139]	Kir6.2, the pore-forming subunit of ATP-sensitive K^+^ channels, is overexpressed in human posttraumatic brain contusions	Contusional TBI	32 patients	NA	KIR6.2 h in contusional astrocytes
Gerzanich et al., 2019 [29]	Sulfonylurea receptor 1, transient receptor potential cation channel subfamily M member 4, and kir6.2: role in hemorrhagic progression of contusion	Specimens from patients with non-ballistic, closed head injury or contusion-TBI who underwent decompressive craniectomy	16 patients	NA	SUR1-TRPM4 co-localization and co-assembly was observed in both microvessels and microglia in the GFAP^−^ contusion core KIR6.2 expressed in CD68^+^ cells but not co-localize with SUR1 No neuronal SUR1-TRPM4 expression in human contusion was reported
Jha et al., 2019 [122]	Downstream *TRPM4* polymorphisms are associated with intracranial hypertension and statistically interact with *ABCC8* polymorphisms in a prospective cohort of severe traumatic brain injury	Candidate gene study in severe TBI (*TRPM4*)	385 patients analyzed	NA	rs8104571 and rs150391806 were associated with intracranial hypertension rs8104571 significantly interacted with *ABCC8* SNPs to moderate effect on intracranial hypertension SNPs spatially clustered, flanking the gene region encoding the channel pore and interfacing with SUR1
Zusman et al., 2021 [34]	Cerebrospinal fluid sulfonylurea receptor-1 is associated with intracranial pressure and outcome after pediatric TBI: an exploratory analysis of the cool kids trial	CSF samples from pediatric patients with severe TBI	16 patients, 7 controls	NA	SUR1 was h in CSF of 9 out of 16 TBI patients and its elevation was associated with increased ICP over 7 days In patients with measurable SUR1 after 24 h, ICP values were high with worse functional outcome at 12 months
Jha et al., 2021 [131]	Genetic variants associated with intraparenchymal hemorrhage progression after traumatic brain injury.	Candidate gene study of hemorrhage progression in severe TBI (*ABCC8*, *TRPM4*)	321 patients analyzed	NA	Four *ABCC8* single nucleotide variants (rs2237982, rs2283261, rs3819521, rs819265) were associated with h hemorrhage progression Four *TRPM4* single nucleotide variants (rs3760666, rs1477363, rs10410857, rs909010) were associated with i hemorrhage progression All SNPs were brain specific expression quantitative trait loci (eQTL) Regulatory annotations revealed promoter and enhancer marks and strong and/or active brain-tissue transcription, and eQTLs directionally and biologically consistent with h progression for *ABCC8* and i expression for *TRPM4*.
Clinical trials: TBI					
Zafardoost et al., 2016 [42]	Evaluation of the effect of glibenclamide in patients with diffuse axonal injury due to moderate to severe head trauma	Diffuse axonal injury Randomized trial	40 patients	Oral GLI 1.25 mg every 12 h for 1 week	Improved Glasgow Coma Scale (GCS) scores and Glasgow Outcome Scale scores
Khalili et al., 2017 [138]	Effects of oral glibenclamide on brain contusion volume and functional outcome of patients with moderate and severe traumatic brain injuries: a randomized double-blind placebo-controlled clinical trial	Moderate to severe contusional TBI Randomized trial	66 patients	Oral GLI 10 mg daily for 10 d	Lower contusion expansion ratios between baseline–day 3 and baseline–day 7
Eisenberg et al., 2020 [31]	Magnetic resonance imaging pilot study of intravenous glyburide in traumatic brain injury	TBI with GCS 4-14 Randomized Trial	28 patients	GLI total daily dose on D1 was 3.12 mg, on D2 and D3 was 2.67 mg/day	Lesion volume increased 1036% with placebo and was 136% with GLI (not significant) 14 patients had contusional TBI

**Table 5 ijms-22-11899-t005:** Pre-clinical and clinical studies evaluating SUR1 inhibition for spinal cord injury (SCI).

Authors, Year	Study Title	Model	Species	GLI Dose	Results
In vivo studies: SCI					
Simard et al., 2007 [28]	Endothelial sulfonylurea receptor 1-regulated NC Ca-ATP channels mediate progressive hemorrhagic necrosis following spinal cord injury	Severe unilateral SCI (10 g weight dropped from 2.5 cm)	Rat	GLI Time: 2-3 min after SCI Dose: 200 ng/h Others: repaglinide, *Abcc8* antisense oligo deoxynucleotide ODN	PHN associated with upregulation of SUR1 in time dependent manner GLI i PHN, improved lesion volume and neurobehavioral assessment
Gerzanich et al., 2009 [139]	De novo expression of Trpm4 initiates secondary hemorrhage in spinal cord injury	Unilateral SCI	Mouse Rat	*Trpm4^−/−^* *Trpm4* antisense ODN	i secondary hemorrhage, lesion expansion, prevented PHN, resulted in minimal capillary fragmentation and improved functional outcome
Simard et al., 2010 [27]	Brief suppression of *Abcc8* prevents autodestruction of spinal cord after trauma	Unilateral SCI	Mouse Rat	GLI Time: 15 min after SCI Loading dose: 10 mg/kg Infusion rate: 200 ng/h *Abcc8^−/−^* *Abcc8* antisense ODN	i secondary hemorrhage, lesion expansion, prevented PHN, resulted in minimal capillary fragmentation and improved functional outcome
Simard et al., 2012 [148]	Spinal cord injury with unilateral versus bilateral primary hemorrhage–effects of glibenclamide	Unilateral/Bilateral primary SCI	Rat	GLI Time: min after SCI Loading dose: 10 mg/kg Infusion rate: 200 ng/h	GLI improved functional benefit as early as 24 h, and i 6-week lesion volume
Simard et al., 2012 [149]	Comparative effects of glibenclamide and riluzole in a rat model of severe cervical spinal cord injury	Unilateral cervical SCI	Rat	GLI Time: 3 h after SCI Loading dose:10 mg/kg Infusion rate: 200 ng/h Other: riluzole	GLI improved capillary fragmentation, PHN, Basso, Beattie and Bresnahan locomotor scores and mortality GLI outperformed riluzole
Hosier et al., 2015 [147]	A direct comparison of three clinically relevant treatments in a rat model of cervical spinal cord injury	Unilateral impact to the cervical spinal cord at C7	Rat	GLI Time: 4 h after SCI Loading dose: 10 mg/kg, Infusion rate: 400 ng/h Others: hypothermia, riluzole	i mortality with hypothermia i Lesion volume with maximal benefit seen in GLI treated rats vs. the other treatment groups
Yao et al., 2018 [153]	Flufenamic acid inhibits secondary hemorrhage and BSCB disruption after spinal cord injury	Thoracic SCI	Mouse	Flufenamic acid	Flufenamic acid inhibited TRPM4 expression, downregulated MMP-2, MMP-9 at 24 h after SCI FFA protected motor neurons and improved locomotor function
Human studies: SCI					
Simard et al., 2010 [27]	Brief suppression of *Abcc8* prevents autodestruction of spinal cord after trauma	Autopsy sample from patients with traumatic SCI	Human	NA	Prominent penumbral SUR1 expression SUR1 in white matter and neurons

**Table 6 ijms-22-11899-t006:** Pre-clinical and clinical studies evaluating SUR1 inhibition for subarachnoid hemorrhage (SAH).

Authors, Year	Study Title	Model	Species	GLI Dose	Results
In vivo studies: SAH					
Simard et al., 2009 [40]	Glibenclamide reduces inflammation, vasogenic edema, and caspase-3 activation after subarachnoid hemorrhage	Mild-moderate SAH using a filament-based endovascular puncture of internal carotid artery	Rat	GLI Time: <15 min after SAH Loading dose: 10 mg/kg Infusion rate: 200 ng/h	SUR1 h at 24 h in neurons and microvascular cells GLI treatment i BBB disruption, local inflammation and reactive astrogliosis, minimal no caspase-3 activation
Tosun et al., 2013 [36]	Inhibition of the Sur1-Trpm4 channel reduces neuroinflammation and cognitive impairment in subarachnoid hemorrhage	SAH induced by stereotactic injection of fresh non-heparinized autologous blood into the subarachnoid space of the entorhinal cortex, OR by filament puncture of carotid artery	Rat	GLI Time: <10 min after SAH Loading dose: 10 mg/kg Infusion rate: 200 ng/h	i BBB disruption, TNFα expression, improved functional score including platform search strategies and rapid spatial learning tasks
Fang et al., 2020 [110]	Pituitary adenylate cyclase-activating polypeptide attenuates brain edema by protecting blood-brain barrier and glymphatic system after subarachnoid hemorrhage in rats	SAH	Rat	PACAP38	Preserved BBB function, accelerated CSF movement attenuated 24 h brain edema and improved neurological score h SUR1 expression associated with MMP-9 elevation and ZO-1 reduction
Human studies: SAH					
Tosun et al., 2013 [36]	Inhibition of the Sur1-Trpm4 channel reduces neuroinflammation and cognitive impairment in subarachnoid hemorrhage	Autopsy sample from SAH patients	7 patients	NA	Prominent expression of SUR1-TRPM4 in neurons, microvessels, astrocytes
Dundar et al., 2020 [37]	Serum SUR1 and TRPM4 in patients with subarachnoid hemorrhage	Aneurysmal SAH patients	44 patients	NA	Elevated serum SUR1 and TRPM4 levels on D1, D4 and D14 post-bleed, associated with GOS

**Table 7 ijms-22-11899-t007:** Pre-clinical and clinical studies evaluating SUR1 inhibition for other neurological diseases.

Authors, Year	Study Title	Model	Species	GLI Dose	Results
In vivo studies: Cardiac arrest					
Huang et al., 2015 [169]	Glibenclamide improves survival and neurologic outcome after cardiac arrest in rats	8 min asphyxia cardiac arrest	Rat	GLI Time: 10 min Loading dose:10 μg/kg Maintenance dose: 1.2 μg at 6, 12, 18, 24 h	h *Abcc8* and *Trpm4* mRNA and SUR1/TRPM4 protein Increased 7-day survival, reduced neurological deficit scores and reduced hippocampal neuronal loss
Huang et al., 2016 [170]	Glibenclamide is comparable to target temperature management in improving survival and neurological outcome after asphyxial cardiac arrest in rats	10 min asphyxia cardiac arrest	Rat	GLI Time: at randomization Loading dose: 10 μg/kg 4 maintenance doses of 1.2 μg per 6 h after ROSC	h *Abcc8* and *Trpm4* mRNA and SUR1/TRPM4 protein Reduced neuronal injury and neuronal deficit
Huang et al., 2018 [171]	Glibenclamide prevents water diffusion abnormality in the brain after cardiac arrest in rats	15 min asphyxia cardiac arrest	Rat	GLI Time: 15 min post-ROSC Loading dose 10 μg/kg Maintenance: four maintenance doses of 1.2 μg at 6, 12, 18, and 24 h after ROSC	Abnormal diffusion restriction was alleviated by GLI treatment within 72 h Reduced neuronal injury and neuronal deficit
Nakayama et al., 2018 [168]	Glibenclamide and therapeutic hypothermia have comparable effect on attenuating global cerebral edema following experimental cardiac arrest	CA induced by IV injection of 0.05 mL cold 0.5 mol/L KCL	Mouse	GLI Loading dose: 10 mg/kg Maintenance dose: 4 mg/kg 6 h and 18 h post ROSC	*Abcc8* and *Trpm4* mRNA5 at 6 h post cardiopulmonary resuscitation i TNFα, IL-6 and NFκB
In vivo studies: Intracerebral hemorrhage (ICH)					
Tosun et al., 2013 [160]	The protective effect of glibenclamide in a model of hemorrhagic encephalopathy of prematurity	Model of hemorrhagic encephalopathy induced by 20 min of intrauterine ischemia, followed by an intraperitoneal injection of glycerol	Rat	GLI Time: immediately before closing the laparotomy Loading dose:10 μg/kg Infusion rate 400 ng/h	SUR1 h in periventricular tissue by 24 h after intrauterine ischemia SUR1 h was also observed in choroid plexus, ependymal lining of lateral ventricles, internal capsule, subventricular zone, corpus callosum and hippocampus
Jiang et al., 2017 [172]	Role of glibenclamide in brain injury after intracerebral hemorrhage	Autologous blood infusion ICH model	Rat	GLI Time: end of surgery Loading dose: 10 μg/kg, Infusion rate: 200 ng/h	SUR1 h in perihematomal neurons and endothelial cells GLI treatment i brain water content, MMP expression and restored BBB integrity
Zhou et al., 2018 [175]	Neuroprotective potential of glibenclamide is mediated by antioxidant and anti-apoptotic pathways in intracerebral hemorrhage	Collagenase induced ICH model	Rat	GLI Time: 30 min before surgery Loading dose:10 μg/kg i.p Maintenance: 1 mg/kg daily	GLI regulated iNOS level in microglia, h BCL-2/BAX ratio, i perihematomal caspase-3 expression and apoptosis
Xu et al., 2019 [143]	Glibenclamide ameliorates the disrupted blood–brain barrier in experimental intracerebral hemorrhage by inhibiting the activation of NLRP3 inflammasome	Autologous blood infusion ICH model	Mouse	GLI Time: immediately after ICH Dose:10 μg/kg i.p.	GLI i perihematomal levels of cytokines, IL1β, IL18, IL6, TNFα, reduced apoptotic cells
Wilkinson et al., 2019 [174]	Glibenclamide, a Sur1-Trpm4 antagonist, does not improve outcome after collagenase-induced intracerebral hemorrhage	Collagenase induced ICH model	Rat	GLI Time: 2 h post-ICH Loading dose: 10 μg/kg Infusion rate: 200 ng/h	GLI did not improve edema, BBB integrity, lesion volume GLI did not improved neurological impairment
Kung et al., 2021 [173]	Glibenclamide does not improve outcome following severe collagenase-induced intracerebral hemorrhage in rats	Collagenase induced ICH model	Rat	GLI Time: 2 h post-ICH Loading dose 10 μg/kg Infusion rate: 200 ng/h	GLI did not improve edema, hematoma volume, or functional outcome at any timepoint
Human studies: ICH					
Simard et al., 2008 [177]	Sulfonylurea receptor 1 in the germinal matrix of premature infants	Brain specimen from pre-mature infants with germinal matrix hemorrhage	12 patients	NA	h SUR1 and HIF1α expression
In vivo studies: Multiple sclerosis (MS) and experimental autoimmune encephalitis (EAE)					
Schattling et al., 2012 [178]	TRPM4 cation channel mediates axonal and neuronal degeneration in experimental autoimmune encephalomyelitis and multiple sclerosis	EAE was induced using myelin oligodendrocyte glycoprotein 35–55	Mouse	GLI Dose: 10 μg daily i.p	GLI i axonal and neuronal degeneration
Makar et al., 2015 [97]	Silencing of *Abcc8* or inhibition of newly upregulated Sur1-Trpm4 reduce inflammation and disease progression in experimental autoimmune encephalomyelitis	EAE was induced in wild-type (WT) and *Abcc8*^−/−^ mice using myelin oligodendrocyte glycoprotein 35–55	Mouse	GLI Dose: 10 μg daily i.p	GLI treatment showed fewer invading immune cells expressing CD45, CD3, CD20 and CD11b GLI also i cells for pro inflammatory cytokines TNFα, Il-7 and INF-γ
Gerzanich et al., 2017 [98]	Salutary effects of glibenclamide during the chronic phase of murine experimental autoimmune encephalomyelitis.	EAE was induced using myelin oligodendrocyte glycoprotein 35–55	Mouse	GLI Dose: 10 μg glibenclamide daily i.p	GLI improved clinical scores, reduced myelin loss, inflammation (CD45, CD20, CD3, p65), and reactive astrocytosis i TNF, BAFF, CCL2 and NOS2 in lumbar spinal cord white matter
Human studies: MS and EAE					
Gerzanich et al., 2017 [98]	Salutary effects of glibenclamide during the chronic phase of murine experimental autoimmune encephalomyelitis.	Demyelinating lesions from MS patients	9 patients	NA	h SUR1 expression in astrocyte like GFAP+ cells Co-expression of SUR1-TRPM4 in stellate cells observed in perivascular endfeet

## Data Availability

Not applicable.

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
