# Peer review of "Sulfonylurea Receptor 1 in Central Nervous System Injury: An Updated Review"

_ijms, 2021, doi:10.3390/ijms222111899_

Round 1

Reviewer 1 Report

Ms. No.: IJMS-1398571
Title: Sulfonylurea Receptor 1 in Central Nervous System Injury: an Updated Review

Reviewer comments

An interesting, well-designed, and well-written “updated” review focuses on the role of the Sulfonylurea Receptor 1 (SUR1) in various CNS injury models. The importance of the SUR1 receptor can’t be questioned, as indicated by a large body of knowledge accumulated to date. This review is a worthy contribution since it summarizes all significant discoveries made over the years. It is a concise almanac of various pathways (cerebral edema, BBB permeability, neuroinflammation) where the SUR1 plays a pivotal role. Pre-clinical models that include the application of drugs (glibenclamide) and their clinical trials counterparts are described in sufficient detail and cover the most important findings to date.

Specific comments

Page 5, line 43: “In the context of a rigid skull, by increasing brain-tissue volume, cerebral edema often lowers brain-tissue perfusion, thereby causing cellular distress/injury. This also increases the risk of herniation and death (10).” Does that mean a limited supply of oxygen and electrolytes to the brain leads to the injury? What about a mechanical insult to sensitive cellular CNS components resulting from increased intracranial pressure (ICP)? The increased ICP is usually relieved by craniotomy in patients suffering from cerebral edema.

Page 9, line 22: “These data, combined with earlier work in cerebral ischemia (57), confirm the crucial role of Sp1 in the regulation of Abcc8 expression across species.” This sentence might contain an overstatement regarding the role of Sp1 among different species (supported by only a single reference). Was the omnipresence of regulatory role Sp1 described in various injury models, or only in rodents and in vitro?

Page 10, line 12: “Table 1. Key SUR1 signaling pathways.” The authors should consider including the animal model as an additional category in the table (or cell line and their origin, in the case of in vitro studies). That information could be included in the “Disease” column, which might be amended as “Disease model details”, or similar (to mirror Tables 2-5).

Page20, line 6: “Given that SUR1-TRPM4 is upregulated after CNS injury, mechanistically it should be less likely that pre-treatment with glibenclamide would confer significant benefit.” Critical observation. Was any study done on the bioavailability of the glibenclamide to estimate the potential for such pre-treatment success?

Author Response

Comments to the Author

An interesting, well-designed, and well-written “updated” review focuses on the role of the Sulfonylurea Receptor 1 (SUR1) in various CNS injury models. The importance of the SUR1 receptor can’t be questioned, as indicated by a large body of knowledge accumulated to date. This review is a worthy contribution since it summarizes all significant discoveries made over the years. It is a concise almanac of various pathways (cerebral edema, BBB permeability, neuroinflammation) where the SUR1 plays a pivotal role. Pre-clinical models that include the application of drugs (glibenclamide) and their clinical trials counterparts are described in sufficient detail and cover the most important findings to date.

We thank the reviewer for this kind appraisal.

Specific comments

Page 5, line 43: “In the context of a rigid skull, by increasing brain-tissue volume, cerebral edema often lowers brain-tissue perfusion, thereby causing cellular distress/injury. This also increases the risk of herniation and death (10).” Does that mean a limited supply of oxygen and electrolytes to the brain leads to the injury? What about a mechanical insult to sensitive cellular CNS components resulting from increased intracranial pressure (ICP)? The increased ICP is usually relieved by craniotomy in patients suffering from cerebral edema.

This is an important observation and we have expanded this section to discuss further the secondary injury and consequences of raised intracranial pressure as well as the morbidity associated with craniotomy or craniectomy.

In the context of a rigid skull, by increasing brain-tissue volume, cerebral edema often lowers brain-tissue perfusion, thereby causing cellular distress/injury. This also increases the risk of herniation and death (10). This relationship between intracranial pressure (ICP) and cerebral edema was first described in 1783 by the Scottish Surgeon, Alexander Monro and is now well known as the Monro-Kellie hypothesis(15). The increased pressure can also limit cerebral perfusion, and further perpetuate secondary injury. Mechanistically, oncotic edema of endothelial cells and breakdown of the blood-brain barrier (BBB) further contribute to other forms of secondary injury such as hemorrhage progression (16). Currently, cerebral edema and resultant intracranial hypertension are treated with non-specific therapies such as hyperosmolar agents and, when severe, decompressive craniectomy. Although these therapies can be life-saving, they do not always improve functional outcome, nor do they target or prevent underlying pathobiological mechanisms(17–20). Advances in precision medicine have led to an increasing recognition of the fact that a ‘one-size -fits-all’ approach is likely suboptimal(20–23), and the potential benefits of a targeted approach.

 Page 9, line 22: “These data, combined with earlier work in cerebral ischemia (57), confirm the crucial role of Sp1 in the regulation of Abcc8 expression across species.” This sentence might contain an overstatement regarding the role of Sp1 among different species (supported by only a single reference). Was the omnipresence of regulatory role Sp1 described in various injury models, or only in rodents and in vitro?

This is a good point, we have replaced the word ‘confirm’ with ‘suggest’ and added additional references including the involvement of Sp1 in Abcc8 transcription in mice, humans and rats.  

(86). Sp1, in turn, has been shown to induce Abcc8 transcription in mice, humans, and rats (81,82,86). These data, combined with earlier work in cerebral ischemia (65), suggest the crucial role of Sp1 in the regulation of Abcc8 expression across species(81,82,86).

 Page 10, line 12: “Table 1. Key SUR1 signaling pathways.” The authors should consider including the animal model as an additional category in the table (or cell line and their origin, in the case of in vitro studies). That information could be included in the “Disease” column, which might be amended as “Disease model details”, or similar (to mirror Tables 2-5).

Thank you for this suggestion- we have added this information and renamed the column “Disease and Model Details”.

 Page20, line 6: “Given that SUR1-TRPM4 is upregulated after CNS injury, mechanistically it should be less likely that pre-treatment with glibenclamide would confer significant benefit.” Critical observation. Was any study done on the bioavailability of the glibenclamide to estimate the potential for such pre-treatment success?

Interesting point- unfortunately the studies did not look at the bioavailability of glibenclamide to estimate potential for pretreatment success, and we have added this limitation to the text.

Reviewer 2 Report

This is an exciting review of the role of SUR1-TTRPM4 as a mediator in CNS injury. After a careful search, I can confirm, since 2012, there has not even published an in-depth study on this adenosine triphosphate' function. Therefore, this review offers essential updates. The authors have drawn up the review very well; even though it is made up of 62 pages, it is not vague and long-winded (as expected frankly) with many cross-appropriate and recent references making it not difficult to understand. Figures 1 and 3 support a lot the review's understanding. After a careful reading, I did not find any particular drawbacks or shortcomings that required majors revisions.

In conclusion, this review is ready for the publication.

Author Response

Comments to the Author

This is an exciting review of the role of SUR1-TTRPM4 as a mediator in CNS injury. After a careful search, I can confirm, since 2012, there has not even published an in-depth study on this adenosine triphosphate' function. Therefore, this review offers essential updates. The authors have drawn up the review very well; even though it is made up of 62 pages, it is not vague and long-winded (as expected frankly) with many cross-appropriate and recent references making it not difficult to understand. Figures 1 and 3 support a lot the review's understanding. After a careful reading, I did not find any particular drawbacks or shortcomings that required majors revisions.

In conclusion, this review is ready for the publication.

We are very grateful to the reviewer for this kind appraisal.